# ACTIVATION GRADIENT BASED POISONED SAMPLE DETECTION AGAINST BACKDOOR ATTACKS

**Danni Yuan**[1][*] **Mingda Zhang**[1][*]**, Shaokui Wei**[1]**, Li Liu**[2]**, Baoyuan Wu**[1][†]

[1]School of Data Science, The Chinese University of Hong Kong, Shenzhen, Guangdong, 518172, P.R. China
[2] The Hong Kong University of Science and Technology (Guangzhou)
`{danniyuan, mingdazhang, shaokuiwei}@link.cuhk.edu.cn`
`avrillliu@hkust-gz.edu.cn` `wubaoyuan@cuhk.edu.cn`

## ABSTRACT

This work studies the task of poisoned sample detection for defending against data poisoning based backdoor attacks. Its core challenge is finding a generalizable and discriminative metric to distinguish between clean and various types of poisoned samples (*e.g.,* various triggers, various poisoning ratios). Inspired by a common phenomenon in backdoor attacks that the backdoored model tend to map significantly different poisoned and clean samples within the target class to similar activation areas, we introduce a novel perspective of the circular distribution of the gradients *w.r.t.* sample activation, dubbed *gradient circular distribution* (GCD). And, we find two interesting observations based on GCD. One is that the GCD of samples in the target class is much more dispersed than that in the clean class. The other is that in the GCD of target class, poisoned and clean samples are clearly separated. Inspired by above two observations, we develop an innovative three-stage poisoned sample detection approach, called *Activation Gradient based Poisoned sample Detection (AGPD)*. First, we calculate GCDs of all classes from the model trained on the untrustworthy dataset. Then, we identify the target class(es) based on the difference on GCD dispersion between target and clean classes. Last, we filter out poisoned samples within the identified target class(es) based on the clear separation between poisoned and clean samples. Extensive experiments under various settings of backdoor attacks demonstrate the superior detection performance of the proposed method to existing poisoned detection approaches according to sample activation-based metrics. Codes are available at https://github.com/SCLBD/BackdoorBench (PyTorch)

## 1 INTRODUCTION

It is well known that deep neural networks (DNNs) are vulnerable to backdoor attacks (Wu et al., 2023), where the adversary could inject a particular backdoor into the DNN model through manipulating the training dataset or training process. Consequently,the backdoored model will produce a target label when encountering a particular trigger pattern, leading to unexpected security threats in practice. Protecting DNNs from backdoor attacks is an urgent and important task.

Here we focus on defending against the data-poisoning based backdoor attacks by filtering out the potential poisoned samples from a untrustworthy training dataset, *i.e., poisoned sample detection (PSD)*. One of the main challenges for PSD is the information lack of the potential poisoned samples, such as the trigger type, the target class(es), the number of poisoned samples, *etc*. Some seminal works have been developed by exploring some discriminative metrics based on the intermediate activation or predictions of poisoned and clean samples in the backdoored model trained on the untrustworthy dataset, such as activation clustering (AC) (Ma et al., 2023a), STRIP (Gao et al., 2019), SCAn (Tang et al., 2021). However, the assumption that poisoned and clean samples can be distinctly separated in activation space has been challenged in some recent backdoor attacks (Qi et al., 2023a).

---

[*]Equal contribution
[†]Corresponding Author

In this work, we introduce a novel perspective that distinguishes the behavior of poisoned and clean samples by tracking their activation gradients (*i.e.,* the gradient *w.r.t.* activation). It is inspired by the phenomenon that a backdoored model tends to map both poisoned and clean samples within the target class to similar areas in its activation space (Huang et al., 2022), such that they can be predicted as the same label. Considering the significant discrepancy between poisoned and clean samples in their original input space, *their mapping directions should be significantly different*, while the mapping direction could be reflected by the activation gradient direction. Thus, we define a new concept called *gradient circular distribution (GCD)* (introduced in Sec. 3), to capture the distribution of activation gradient directions. Take Fig. 1 as the example, given a trained model, we calculate one GCD of training samples in each class. There are **two interesting observations**:

- **Observation 1 on GCD dispersion**: Given one backdoored model (see middle/right sub-figures), the target GCD is much more **dispersed** than GCDs of all clean classes.

- **Observation 2 on sample separation in target GCD**: In the GCD of target class, poisoned and clean samples are clearly **separated** (see the **black** and **blue** arcs in middle/right sub-figures), and they locate at two **separated** clusters.

Motivated by above two observations, we develop an innovative poisoned sample detection approach, called **Activation Gradient based Poisoned sample Detection (AGPD)**, which consist of three stages. **First**, we train a DNN model based on the untrustworthy dataset, and calculate GCDs of all classes. **Second**, we identify the target class(es) according to a novel class-level metric that measures the dispersion of each class's GCD (corresponding to the first observation). **Last**, within the identified target class(es), we gradually filter out poisoned samples according to a novel sample-level metric that measures the closeness to the clean reference sample (corresponding to the second observation). Moreover, we conduct extensive evaluations under various backdoor attacks and various datasets, and show that the activation gradient is more discriminative than the activation to distinguish between poisoned and clean samples, which explains the superior.

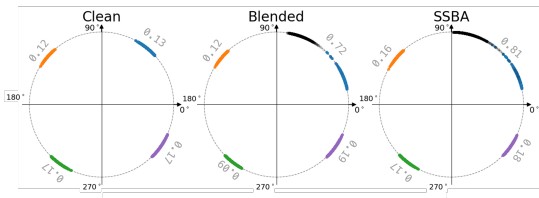

Figure 1: Gradient circular distributions (GCDs) across four classes of CIFAR-10, on the clean model **(left)**, Blended attacked model **(middle)**, and SSBA attacked model **(right)**, respectively. The value along with each arc indicates the CVBT value. The GCD of the target class (covering both **black** and **blue** arcs). Note that we moved three clean classes' arcs to different quadrants to avoid visual overlap.

In summary, the **main contributions** of this work are three-fold. **(1)** We introduce a novel perspective for poisoned sample detection, called gradient circular distribution (GCD), and present two interesting observations based on GCD. **(2)** We develop an innovative approach by sequentially identifying the target class(es) and filtering poisoned samples for the poisoned sample detection task, based on GCD and two novel metrics about GCD. **(3)** We conduct extensive evaluations and analysis to verify the superiority of the proposed approach to existing activation-based detection approaches.

## 2 RELATED WORK

**Backdoor attack.** BadNets (Gu et al., 2019) is the pioneering work that introduces the concept of backdoor attack into Deep Neural Networks (DNNs), in which the adversary manipulates training samples by adding a small patch with specific patterns and changing their labels to a target label. Following this, the variety of triggers expanded significantly, including a cartoon image used in Blended (Chen et al., 2017), a universal adversarial perturbation with only low-frequency components utilized in Low-Frequency (Zeng et al., 2021), and a sinusoidal signal employed in SIG (Barni et al., 2019), *etc*, which use same trigger across different poisoned samples. Sample-specific triggers have been designed, such as WaNet (Nguyen & Tran, 2021), Input-Aware (Nguyen & Tran, 2020), SSBA (Li et al., 2021b), CTRL (Li et al., 2023), TaCT (Tang et al., 2021), and Adap-Blend (Qi et al., 2023a). These attacks use more complex and dynamic triggers, posing significant challenges for poisoned sample detection. Additionally, some attacks explore various attack settings regarding the number of triggers and target classes, such as all-to-all attack (*e.g.,* BadNets-A2A (Gu et al., 2019)), multi-target and multi-trigger attack (*e.g.,* c-BaN (Salem et al., 2022)). These diverse settings further complicate the detection of poisoned samples.

**Backdoor defense.** According to the accessible information, several different branches of backdoor defense methods have been developed, such as the pre-training backdoor defense (*e.g.,* (Ma et al., 2023a; Tran et al., 2018; Al Kader Hammoud et al., 2023)) if given a untrustworthy training dataset, in-training backdoor defense (*e.g.,* (Huang et al., 2022; Li et al., 2021a; Chen et al., 2022; Mu et al., 2023; Gao et al., 2023)) if the training process can be controlled by defender, as well as post-training backdoor defense (*e.g.,* (Liu et al., 2018; Zhu et al., 2023b;a; Wei et al., 2023; Wang et al.; Wu & Wang, 2021; Zeng et al., 2022; Zheng et al., 2022b; Chai & Chen, 2022; Zheng et al., 2022a)) if given a backdoored model. Due to space limitations, we will only review existing methods of poisoned sample detection (PSD), which belongs to pre-training backdoor defense. Most existing PSD methods aim to construct discriminative metrics between poisoned and clean samples based on intermediate activations, final predictions, or loss values. For activation-based methods, such as activation clustering (AC) (Ma et al., 2023a), Beatrix (Ma et al., 2023b), SCAn (Tang et al., 2021), and Spectral (Tran et al., 2018), they utilize dimensionality reduction and clustering techniques, such as K-means clustering, Gram matrix analysis, two-component decomposition, and SVD, to distinguish poisoned and clean samples. For input-based methods, STRIP (Gao et al., 2019) uses the entropy of predictions on perturbed inputs to identify poisoned samples, while CD (Huang et al., 2023) measures the $L_1$ norm of the learned masks on inputs to detect poisoned samples. For loss-based methods, like ABL (Li et al., 2021a) and ASSET (Pan et al., 2023), they observed that the loss of poisoned samples decreases quickly during the early training epochs, leveraging this phenomenon to identify poisoned samples.

## 3 PRELIMINARY: GRADIENT CIRCULAR DISTRIBUTION

**Task setting.** Given a DNN-based classification model $f_{\boldsymbol{w}} : \mathcal{X} \to \mathcal{Y}$, with $\mathcal{X} \in \mathbb{R}^d$ being the input sample space and $\mathcal{Y} = \{1, 2, \ldots, K\}$ being the output space with $K$ candidate classes, as well as a dataset $\mathcal{D} = \{(\boldsymbol{x}_i, y_i)\}_{i=1}^n$, we investigate their gradients of $f_{\boldsymbol{w}}$.

### 3.1 DEFINITION OF GRADIENT CIRCULAR DISTRIBUTION

Here we introduce the definitions of Activation Gradient and Gradient Circular Distribution (GCD), as described in Definition 1 and Definition 2, respectively.

**Definition 1 (Activation Gradient)** *Given a model $f_{\boldsymbol{w}}$, for a sample $\boldsymbol{x}$ labelled as $y$, we denote its activation map at the l-th layer as $\boldsymbol{h}_{\boldsymbol{x}}^{(l)} \in \mathbb{R}^{C^{(l)} \times H^{(l)} \times W^{(l)}}$, where $C^{(l)}$, $H^{(l)}$, $W^{(l)}$ are its depth (number of channels), height and width, respectively. Then, we define the **channel-wise activation gradient $\boldsymbol{g}^{(l)}(\boldsymbol{x}, y) \in \mathbb{R}^{C^{(l)}}$** as*

$$\boldsymbol{g}_{\boldsymbol{w}}^{(l)}(\boldsymbol{x}, y) = \frac{1}{H^{(l)} W^{(l)}} \sum_{h=1}^{H^{(l)}} \sum_{w=1}^{W^{(l)}} \frac{\partial [f_{\boldsymbol{w}}(\boldsymbol{x})]_y}{\partial [\boldsymbol{h}_{\boldsymbol{x}}^{(l)}]_{:,h,w}} \in \mathbb{R}^{C^{(l)}}, \quad (1)$$

*where $[f_{\boldsymbol{w}}(\boldsymbol{x})]_y$ is the logit w.r.t. class $y$ and $[\boldsymbol{h}_{\boldsymbol{x}}^{(l)}]_{:,h,w} \in \mathbb{R}^{C^{(l)}}$ is the activation sliced at height $h$ and width $w$ over all channels. For simplicity, if no special specifications are required, hereafter we will refer to it as $\boldsymbol{g}^{(l)}(\boldsymbol{x})$ for each layer $l$.*

**Definition 2 (Gradient Circular Distribution (GCD))** *Given a model $f_{\boldsymbol{w}}(\cdot)$, a set of samples $\mathcal{D} = \{(\boldsymbol{x}_i, y_i)\}_{i=1}^n$, and a basis sample pair $(\boldsymbol{x}_0, y_0)$, we firstly calculate the activation gradient of each sample for each layer $l$, i.e., $\boldsymbol{g}^{(l)}(\boldsymbol{x}_i) \quad i = 0, 1, \ldots, n$. Then, take $\boldsymbol{g}^{(l)}(\boldsymbol{x}_i)$ as the (unnormalized) basis vector, the angle of each sample in $\mathcal{D}$ is calculated as follows:*

$$\theta_{\boldsymbol{x}_0}^{(l)}(\boldsymbol{x}_i) = \arccos \left( \frac{\boldsymbol{g}^{(l)}(\boldsymbol{x}_i) \cdot \boldsymbol{g}^{(l)}(\boldsymbol{x}_0)}{\|\boldsymbol{g}^{(l)}(\boldsymbol{x}_i)\| \|\boldsymbol{g}^{(l)}(\boldsymbol{x}_0)\|} \right) \in [0, 2\pi), i = 1, \ldots, n, \quad (2)$$

*where $\cdot$ denotes dot product, and $\| \cdot \|$ returns the magnitude. The distribution of the angle set $\{\theta_{\boldsymbol{x}_0}^{(l)}(\boldsymbol{x}_i)\}_{i=1}^n$ is called as the gradient circular distribution (GCD) of $\mathcal{D}$, denoted as $\mathcal{P}_{\boldsymbol{x}_0}^{(l)}(\mathcal{D})$ for each layer $l$.*

### 3.2 CHARACTERISTICS OF GRADIENT CIRCULAR DISTRIBUTION

To accurately capture the characteristics of $\mathcal{P}_{\boldsymbol{x}_0}^{(l)}(\mathcal{D})$ observed in Fig. 1 and Sec. 1, we introduce the following two metrics.

**Dispersion and separation metric of $\mathcal{P}_{\boldsymbol{x}_0}^{(l)}(\mathcal{D})$.** To measure the dispersion and separation of $\mathcal{P}_{\boldsymbol{x}_0}^{(l)}(\mathcal{D})$ for each layer $l$, we design a novel metric called **C**osine similarity **V**ariation towards **B**asis **T**ransition (**CVBT**). Specifically, given $\{\theta_{\boldsymbol{x}_0}^{(l)}(\boldsymbol{x}_i)\}_{i=1}^n$, we firstly pick the activation gradient $\boldsymbol{g}^{(l)}(\boldsymbol{x}_{n^*})$ corresponding to the largest angle, *i.e.*, $n^* = \arg\max_{i \in \{1,...,n\}} \theta_{\boldsymbol{x}_0}^{(l)}(\boldsymbol{x}_i)$. In other words, $\boldsymbol{g}^{(l)}(\boldsymbol{x}_{n^*})$ is the farthest activation gradient vector from the original basis vector $\boldsymbol{g}^{(l)}(\boldsymbol{x}_0)$. Then, by setting $\boldsymbol{g}^{(l)}(\boldsymbol{x}_{n^*})$ as a new basis vector, we calculate $\{\theta_{\boldsymbol{x}_{n^*}}^{(l)}(\boldsymbol{x}_i)\}_{i=1}^n$ using Eq. (2). Based on $\{\theta_{\boldsymbol{x}_0}^{(l)}(\boldsymbol{x}_i)\}_{i=1}^n$ and $\{\theta_{\boldsymbol{x}_{n^*}}^{(l)}(\boldsymbol{x}_i)\}_{i=1}^n$, we formulate the CVBT metric of $\mathcal{P}_{\boldsymbol{x}_0}^{(l)}(\mathcal{D})$ as follows:

$$\rho_{\boldsymbol{x}_0}^{(l)}(\mathcal{D}) = \left( \frac{1}{n} \sum_{i=1}^n \left( \cos(\theta_{\boldsymbol{x}_0}^{(l)}(\boldsymbol{x}_i)) - \cos(\theta_{\boldsymbol{x}_{n^*}}^{(l)}(\boldsymbol{x}_i)) \right)^2 \right)^{\frac{1}{2}} \in [0, 2], \tag{3}$$

where $\cos(\theta)$ returns the cosine value of an angle $\theta$. Note that $\rho_{\boldsymbol{x}_0}^{(l)}(\mathcal{D})$ is positively proportional to dispersion and separation, *i.e.,* larger $\rho_{\boldsymbol{x}_0}^{(l)}(\mathcal{D})$ indicates larger dispersion and larger separation of $\mathcal{P}_{\boldsymbol{x}_0}^{(l)}(\mathcal{D})$. More in-depth analysis will be presented later.

**Sample-level closeness metric based on $\mathcal{P}_{\boldsymbol{x}_0}^{(l)}(\mathcal{D})$** . Given $\{\theta_{\boldsymbol{x}_0}^{(l)}(\boldsymbol{x}_i)\}_{i=1}^n$ and $\{\theta_{\boldsymbol{x}_{n^*}}^{(l)}(\boldsymbol{x}_i)\}_{i=1}^n$, we design a novel metric to measure the closeness of each sample $\boldsymbol{x}_i$ to the reference sample $\boldsymbol{x}_0$, as follows:

$$s_{\boldsymbol{x}_0}^{(l)}(\boldsymbol{x}_i) = \frac{1 - \cos(\theta_{\boldsymbol{x}_{n^*}}^{(l)}(\boldsymbol{x}_i))}{(1 - \cos(\theta_{\boldsymbol{x}_{n^*}}^{(l)}(\boldsymbol{x}_i))) + (1 - \cos(\theta_{\boldsymbol{x}_0}^{(l)}(\boldsymbol{x}_i)))} \in [0, 1). \tag{4}$$

Note that **larger $s_{\boldsymbol{x}_0}^{(l)}(\boldsymbol{x}_i)$ indicates greater closeness of $\boldsymbol{x}_i$ to $\boldsymbol{x}_0$.** For example, if $\boldsymbol{g}^{(l)}(\boldsymbol{x}_i)$ has the same direction with $\boldsymbol{g}^{(l)}(\boldsymbol{x}_{n^*})$ while the opposite direction with $\boldsymbol{g}^{(l)}(\boldsymbol{x}_0)$, then $s_{\boldsymbol{x}_0}^{(l)}(\boldsymbol{x}_i) = 0$, implying the farthest from $\boldsymbol{x}_0$. In contrast, if $\boldsymbol{g}^{(l)}(\boldsymbol{x}_i)$ has the opposite direction with $\boldsymbol{g}^{(l)}(\boldsymbol{x}_{n^*})$ while the same direction with $\boldsymbol{g}^{(l)}(\boldsymbol{x}_0)$, then $s_{\boldsymbol{x}_0}^{(l)}(\boldsymbol{x}_i) = 1$, implying the closest to $\boldsymbol{x}_0$.

**Remark.** Note that the single basis vector $\boldsymbol{g}^{(l)}(\boldsymbol{x}_{n^*})$ in above two metrics could be extended to be a set of basis vectors, *i.e.,* $\mathcal{G}_m = \{\boldsymbol{g}^{(l)}(\boldsymbol{x}_{n_j^*})\}_{j=1,...,m}$, by picking the activation gradients of top-$m$ largest angles among $\{\theta_{\boldsymbol{x}_0}^{(l)}(\boldsymbol{x}_i)\}_{i=1,...,n}$. Correspondingly, above two metrics are adjusted by replacing each basis vector to the original metrics, then calculating the average. This extension's effect will be analyzed in later evaluations about adaptive attacks.

## 4 ACTIVATION GRADIENT BASED POISONED DETECTION METHOD

### 4.1 PROBLEM SETTING

**Threat model.** We consider the threat model of data poisoning based backdoor attack. The adversary generates a poisoned dataset $\mathcal{D}_{bd}$, containing a clean subset $\mathcal{D}_c = \{(\boldsymbol{x}_i, y_i)\}_{i=1}^{n_c}$ and a poisoned subset $\mathcal{D}_p = \{(\tilde{\boldsymbol{x}}_i, t)\}_{i=1}^{n_p}$. $\boldsymbol{x}, \tilde{\boldsymbol{x}} \in \mathcal{X}$ denotes the clean and poisoned sample with trigger, respectively. $y, t \in \mathcal{Y}$ indicates the ground-truth and target label, respectively. We denote $r = \frac{n_p}{n_c + n_p}$ as the poisoning ratio. Note that there could be multiple triggers (*i.e.,* multi-trigger) and multiple target labels (*i.e.,* multi-target) in the poisoned subset.

**Defender's goal.** The defender aims to identify poisoned samples from the untrustworthy dataset $\mathcal{D}_{bd}$. We assume that the defender has access to $\mathcal{D}_{bd}$, and a small set of additional clean samples $\mathcal{D}_{ac}$, which contains at least one clean sample for each class, as suggested in previous works (Ma et al., 2023b)(Tang et al., 2021)(Gao et al., 2019). Besides, the defender has the capability to train a DNN model $f_{\boldsymbol{w}_{bd}} : \mathcal{X} \to \mathcal{Y}$ based on $\mathcal{D}_{bd}$.

### 4.2 POISONED SAMPLE DETECTION METHOD

Inspired by the two observations demonstrated in Sec. 1 and Fig. 1, we develop an innovative poisoned sample detection method by utilizing GCD and the corresponding metrics (see Sec. 3), called **A**ctivation **G**radient based **P**oisoned **D**etection (**AGPD**). As illustrated in Fig. 2, AGPD consists of three stages, as detailed below.

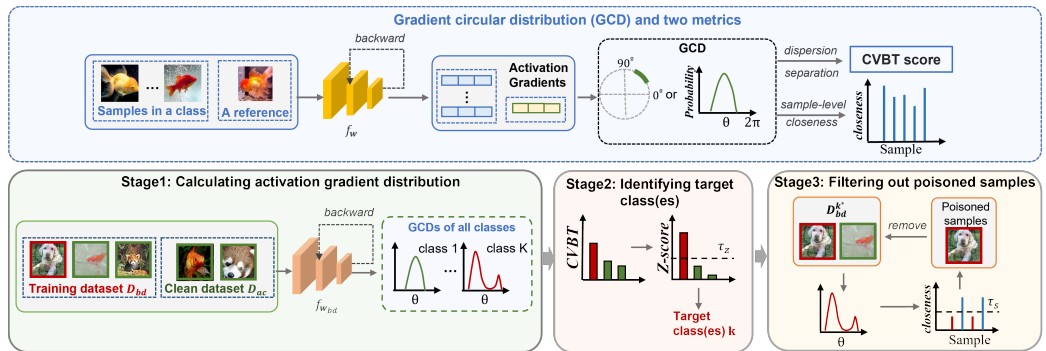

Figure 2: Illustrations of gradient circular distribution (GCD) and two metrics, and the pipeline of the proposed APGD method which consists of three stages: 1) calculating activation gradient distribution, 2) identifying target class(es), and 3) filtering out poisoned samples within the identified target class(es).

**Stage 1: Calculating activation gradient distribution.** We denote the samples of class $k$ in $\mathcal{D}_{bd}$ as $\mathcal{D}_{bd}^k = \{(\boldsymbol{x}_i, k)\}_{i=1}^{n_k}$. Given the model $f_{\boldsymbol{w}_{bd}}$ trained on $\mathcal{D}_{bd}$ (the training details will be provided in Appendix B.2), and picking one clean sample pair $(\boldsymbol{x}_0^k, k) \in \mathcal{D}_{ac}$ as the reference, we can calculate the GCD of $\mathcal{D}_{bd}^k$, according to Eqs. (1) and (2). Consequently, we obtain $\{\mathcal{P}_{\boldsymbol{x}_0^k}^{(l)}(\mathcal{D}_{bd}^k)\}_{l,k=1,1}^{L,K}$. Note that as defined in Eq. (1), the superscript $(l)$ indicates that we adopt the activation gradients of the $l$-th layer in $f_{\boldsymbol{w}_{bd}}$ to calculate GCD. For simplicity, hereafter we denote $\mathcal{P}_{\boldsymbol{x}_0^k}^{(l)}(\mathcal{D}_{bd}^k)$ as $\mathcal{P}_k^{(l)}$.

**Stage 2: Identifying target class(es).** According to the aforementioned first observation, the GCD of the target class is likely to be more dispersed than that of the clean class. Thus, we firstly calculate the dispersion value $\rho_{\boldsymbol{x}_0^k}^{(l)}(\mathcal{D}_{bd}^k)$ (for simplicity, we denote it as $\rho_k^{(l)}$) of each $\mathcal{P}_k^{(l)}$, according to the CVBT metric (see Eq. (3)). As shown in Fig. 2, since $\rho_k^{(l)}$ of target class(es) is likely to be larger, while those of clean classes are likely to be small, we can adopt the anomaly detection technique to identify target class(es), such as the absolute robust Z-score (Iglewicz & Hoaglin, 1993). Specifically, we calculate Z-score of $\rho_k^{(l)}$ as follows:

$$z_k^{(l)} = \frac{\rho_k^{(l)} - \tilde{\rho}^{(l)}}{\gamma \times \mathrm{MAD}(\{\rho_k^{(l)}\}_{k=1}^K)}, \tag{5}$$

where $\mathrm{MAD}(\{\rho_k^{(l)}\}) = \mathrm{median}(\{|\rho_k^{(l)} - \tilde{\rho}^{(l)}|\})$ indicates the median-absolute-deviation (MAD), and $\tilde{\rho}^{(l)}$ denotes the median value of $\{\rho_k^{(l)}\}_{k=1}^K$. $\gamma$ is a statistical constant valued at 1.4826. Larger $z_k^{(l)}$ indicates larger likelihood of anomaly. We firstly choose the layer with the largest Z-score, *i.e.*, $l^* = \arg\max_l(\max_k z_k^{(l)})$. Then, **if $z_k^{(l^*)}$ exceeds a threshold $\tau_z$ (specified later), *i.e.*, $z_k^{(l^*)} \geq \tau_z$, $k$ is identified as a target class**, otherwise clean class.

**Stage 3: Filtering out poisoned samples within the identified target class(es).** Inspired by the second observation mentioned in Sec. 1 that the poisoned sample is likely to be far from the clean sample in GCD, here develop a novel algorithm which gradually filters out poisoned samples with the identified target class(es). Specifically, as illustrated in Fig. 2, when obtained the identified target class $k^*$, we firstly pick one clean sample pair of class $k^*$ from $\mathcal{D}_{ac}$ as the reference $(\boldsymbol{x}_0, k^*)$, then we conduct the following three steps iteratively, until a stopping criteria is satisfied:

1. For the set $\mathcal{D}_{bd}^{k^*}$, we calculate its GCD according to Definition 2, *i.e.*, $\mathcal{P}_{k^*}^{(l^*)}$;

2. We calculate sample-level closeness value $s_{\boldsymbol{x}_0}(\boldsymbol{x}_i)$ for each $\boldsymbol{x}_i \in \mathcal{D}_{bd}^{k^*}$;

3. If the closeness value of one sample is lower than threshold $\tau_s$ (specified in experiments), *i.e.*, $s_{\boldsymbol{x}_0}(\boldsymbol{x}_i) < \tau_s$, then this sample is identified as poisoned, as it is far from the clean reference $\boldsymbol{x}_0$. Then, $\mathcal{D}_{bd}^{k^*}$ is updated by removing these identified poisoned samples.

In terms of the stopping criteria, we propose to firstly conduct the above iterations until $\mathcal{D}_{bd}^{k^*}$ becomes a null set. At each iteration, we calculate the distribution of $\{s_{\boldsymbol{x}_0}(\boldsymbol{x}_i)\}_{\boldsymbol{x}_i \in \mathcal{D}_{bd}^{k^*}}$, and the Jensen–Shannon

Table 1: The detection performance of AGPD and compared detectors on CIFAR-10 and Tiny ImageNet, respectively, with the model Preact-ResNet18.

| Dataset | Attack | No defense ACC/ASR | AC TPR↑ | FPR↓ | F1↑ | Beatrix TPR↑ | FPR↓ | F1↑ | SCAn TPR↑ | FPR↓ | F1↑ | Spectral TPR↑ | FPR↓ | F1↑ | STRIP TPR↑ | FPR↓ | F1↑ | ABL TPR↑ | FPR↓ | F1↑ | CD TPR↑ | FPR↓ | F1↑ | ASSET TPR↑ | FPR↓ | F1↑ | AGPD (Ours) TPR↑ | FPR↓ | F1↑ |
|---|---|---|---|---|---|---|---|---|---|---|---|---|---|---|---|---|---|---|---|---|---|---|---|---|---|---|---|---|---|
| CIFAR-10 | BadNets | 91.82/93.79 | 0.00 | **0.00** | 0.00 | 87.24 | 8.95 | 65.15 | **96.04** | **0.00** | **97.98** | 16.76 | 1.30 | 26.09 | 90.16 | 10.19 | 63.97 | 89.74 | 1.14 | 89.74 | 78.02 | 43.87 | 27.24 | 3.16 | 47.66 | 1.19 | 90.06 | 0.03 | 94.65 |
| | Blended | 93.69/99.75 | 0.00 | **0.00** | 0.00 | 47.60 | 5.07 | 49.28 | 99.62 | **0.00** | 99.81 | 28.04 | 0.05 | 43.64 | 61.42 | 11.31 | 46.67 | 82.14 | 1.98 | 82.14 | 85.06 | 49.62 | 26.93 | 3.70 | 12.66 | 3.40 | **99.98** | 0.02 | **99.88** |
| | LF | 93.01/99.05 | 0.00 | **0.00** | 0.00 | 0.00 | 10.72 | 0.00 | 95.58 | 0.01 | 97.71 | 0.04 | 3.16 | 0.06 | 86.92 | 10.09 | 62.59 | 45.48 | 6.06 | 45.48 | 88.44 | 24.33 | 43.42 | 3.80 | 10.28 | 3.87 | **99.80** | 0.07 | **99.60** |
| | SSBA | 92.88/97.06 | 0.00 | **0.00** | 0.00 | 10.26 | 8.54 | 10.97 | 97.34 | 0.01 | 98.60 | 27.14 | 0.15 | 42.24 | 77.42 | 11.71 | 54.75 | 67.38 | 3.62 | 67.38 | 91.30 | 3.76 | 81.12 | 3.56 | 46.36 | 1.37 | **99.62** | 0.04 | **99.63** |
| | SIG | 93.40/95.43 | 0.00 | **0.00** | 0.00 | 0.00 | **0.00** | 0.00 | 99.52 | **0.00** | 99.74 | 0.00 | 1.58 | 0.00 | 99.44 | 9.68 | 51.88 | 90.80 | 5.75 | 60.53 | 85.88 | 21.28 | 45.51 | 0.48 | 32.73 | 0.13 | **100.0** | 0.04 | 99.66 |
| | CTRL | 95.52/98.8 | 0.00 | 9.92 | 0.00 | 0.00 | 6.26 | 0.00 | 0.00 | 5.00 | 0.00 | 0.40 | 15.77 | 0.20 | 99.80 | 9.47 | 52.57 | 90.32 | 5.77 | 60.21 | 99.44 | 0.55 | 97.31 | 67.88 | 51.60 | 11.82 | 99.76 | 0.01 | 99.78 |
| | WaNet | 89.68/96.94 | 0.00 | 2.77 | 0.00 | 0.92 | 9.74 | 0.94 | 87.39 | 0.07 | 92.96 | 0.90 | 2.97 | 1.38 | 1.22 | 9.13 | 1.28 | 18.92 | 9.08 | 18.31 | 86.88 | 79.77 | 19.20 | 0.55 | 1.22 | 1.01 | **97.80** | 0.31 | **97.40** |
| | Input-Aware | 90.82/98.17 | 0.00 | 3.31 | 0.00 | 0.41 | 10.92 | 0.39 | **99.15** | 0.47 | **97.36** | 1.51 | 2.90 | 2.34 | 0.81 | 9.07 | 0.86 | 0.17 | 11.02 | 0.17 | 82.85 | 18.99 | 46.84 | 2.82 | 60.13 | 0.90 | 88.25 | 1.13 | 88.61 |
| | TaCT | 93.21/95.95 | 0.00 | **0.00** | 0.00 | 75.94 | 19.98 | 42.69 | **100.0** | **0.00** | **99.99** | 29.76 | 0.03 | 45.78 | 67.60 | 8.59 | 55.20 | 33.80 | 7.36 | 33.80 | 80.52 | 53.90 | 24.19 | 7.74 | 59.90 | 2.64 | **100.0** | 0.07 | 99.68 |
| | Adap-Blend | 92.87/66.17 | 0.00 | **0.00** | 0.00 | 4.62 | 8.33 | 5.14 | **99.16** | 1.15 | **94.66** | 24.34 | 0.47 | 37.85 | 14.66 | 11.82 | 13.27 | 0.08 | 11.10 | 0.00 | 0.00 | **0.00** | 0.00 | 97.22 | 39.52 | 37.88 | 89.32 | 0.33 | 92.89 |
| | BadNets-A2A | 91.93/74.40 | 28.76 | 4.51 | 33.96 | 40.28 | 9.25 | 36.04 | 0.00 | **0.00** | 0.00 | 0.00 | 1.67 | 0.00 | 1.80 | 17.08 | 1.41 | 2.48 | 10.84 | 2.48 | 67.20 | 45.72 | 23.23 | 3.32 | 3.31 | 4.99 | **97.32** | 0.02 | **98.57** |
| | SSBA-A2A | 93.46/87.84 | 50.02 | 2.66 | 57.51 | 19.04 | 5.26 | 22.88 | 0.00 | **0.00** | 0.00 | 0.00 | 1.67 | 0.00 | 12.74 | 9.87 | 12.64 | 0.08 | 11.00 | 0.96 | 75.54 | 44.43 | 26.26 | 48.74 | 49.56 | 16.39 | **98.06** | 0.02 | **98.95** |
| | Avg. | | 6.57 | 1.93 | 7.62 | 23.86 | 8.58 | 19.46 | 72.82 | 0.56 | 73.23 | 10.74 | 2.64 | 16.63 | 51.17 | 10.67 | 34.76 | 43.45 | 7.06 | 38.43 | 76.76 | 32.18 | 38.44 | 20.25 | 34.58 | 7.13 | **96.66** | 0.17 | **97.44** |
| Tiny ImageNet | BadNets | 56.12/99.90 | 0.00 | 0.40 | 0.00 | 1.11 | 9.64 | 1.18 | **100.0** | **0.00** | **100.0** | 14.04 | 0.18 | 24.28 | **100.0** | 11.51 | 65.89 | 95.49 | 0.50 | 95.49 | 66.91 | 51.31 | 21.28 | 95.11 | 38.62 | 35.05 | 99.90 | 0.16 | 99.24 |
| | Blended | 55.53/97.57 | 0.00 | 1.26 | 0.00 | 0.53 | 10.06 | 0.55 | 99.85 | **0.00** | 99.92 | 11.45 | 0.47 | 19.80 | 96.51 | 11.89 | 63.59 | 90.18 | 1.09 | 90.18 | 93.29 | 9.94 | 66.00 | 78.53 | 60.74 | 21.66 | **100.0** | 0.05 | **99.78** |
| | LF | 55.21/98.51 | 15.05 | 1.26 | 23.82 | 19.36 | 9.15 | 19.20 | 63.86 | **0.00** | 77.94 | 11.35 | 0.48 | 19.62 | 85.97 | 9.72 | 62.88 | 87.44 | 1.40 | 87.44 | 95.22 | 6.65 | 74.65 | 54.28 | 50.69 | 17.78 | **100.0** | 0.10 | **99.56** |
| | SSBA | 55.97/97.69 | 0.00 | 1.05 | 0.00 | 0.45 | 8.70 | 0.50 | 59.11 | **0.00** | 74.30 | 13.97 | 0.19 | 24.15 | **99.96** | 11.20 | 66.46 | 95.24 | 0.53 | 95.24 | 88.07 | 20.94 | 46.78 | 26.90 | 57.36 | 8.37 | **99.89** | 0.04 | **99.75** |
| | WaNet | 58.33/90.35 | 13.73 | 0.80 | 22.60 | 75.94 | 11.88 | 52.23 | 62.72 | **0.00** | 77.09 | 11.37 | 0.45 | 19.65 | 6.38 | 11.11 | 5.97 | 94.40 | 1.27 | 91.35 | 76.73 | 20.18 | 42.83 | 0.00 | 15.12 | 0.00 | **99.77** | 0.12 | **99.30** |
| | Input-Aware | 57.5/99.75 | 0.00 | 0.68 | 0.00 | 64.97 | 10.30 | 49.13 | 99.65 | **0.00** | 99.82 | 12.92 | 0.29 | 22.32 | 12.02 | 11.08 | 0.97 | 72.52 | 3.53 | 70.18 | 80.22 | 28.93 | 36.41 | 3.07 | 5.97 | 3.97 | **99.89** | 0.04 | **99.73** |
| | TaCT | 54.93/91.25 | 0.00 | 1.02 | 0.00 | 45.51 | 10.13 | 38.45 | **100.0** | **0.00** | **100.0** | 15.48 | 0.03 | 26.75 | 80.03 | 16.37 | 48.90 | 32.25 | 7.53 | 32.25 | 99.58 | 99.45 | 18.19 | 35.77 | 56.31 | 12.20 | 99.99 | 0.43 | 98.11 |
| | Adap-Blend | 54.55/96.35 | 0.00 | 0.82 | 0.00 | 9.56 | 9.39 | 9.85 | 47.24 | 0.05 | 63.98 | 15.14 | 0.06 | 26.18 | 77.73 | 15.85 | 48.52 | 8.97 | 10.11 | 8.97 | 73.69 | 39.65 | 27.77 | 71.95 | 49.92 | 25.19 | **99.96** | 0.17 | **99.24** |
| | Avg. | | 3.60 | 0.91 | 5.80 | 27.18 | 9.91 | 21.39 | 79.05 | 0.01 | 86.63 | 13.21 | 0.27 | 22.84 | 69.83 | 12.34 | 46.65 | 72.06 | 3.24 | 71.39 | 84.21 | 34.63 | 41.74 | 45.70 | 41.84 | 15.53 | **99.92** | 0.14 | **99.34** |

(JS) divergence between the current and its previous distribution. Then, we adopt a trace-back strategy by checking the JS divergence value of all iterations, and the iteration that its JS divergence locates at the stable and low region could be set as the stopping iteration. Due to the space limit, more details of the whole algorithm, as well as the stopping criteria, will be presented in Appendix D.1.

# 5 EXPERIMENTS

## 5.1 EXPERIMENTAL SETUP

**Attack settings.** To evaluate the performance of our detection method, we conduct 10 state-of-the-art (SOTA) backdoor attacks that cover 4 categories: 1) *non-clean label with sample-agnostic trigger*, such as BadNets (Gu et al., 2019), Blended (Chen et al., 2017), LF (Zeng et al., 2021); 2) *clean-label with sample-agnostic trigger*, like SIG (Barni et al., 2019); 3) *clean-label with sample-specific trigger*, such as CTRL (Li et al., 2023), an attack based on self-supervised learning; and 4) *non-clean label with sample-specific trigger*, including SSBA (Li et al., 2021b), WaNet (Nguyen & Tran, 2021), Input-Aware (Nguyen & Tran, 2020), TaCT (Tang et al., 2021), and Adap-Blend (Qi et al., 2023a). These attack settings follow BackdoorBench (Wu et al., 2022) for a fair comparison. The poisoning ratio in our main evaluation is 10% for non-clean label attacks and 5% for clean label attacks. The target label $t$ is set to 0 for all-to-one backdoor attack, while target labels are set to $t = (y + 1) \mod K$ for all-to-all backdoor attack. The detailed experimental setting are provided in Appendix B.3

**Detection settings.** We compare AGPD with eight detection methods, categorized into three groups: 1) *activation-based*, including AC (Ma et al., 2023a), Beatrix (Ma et al., 2023b), SCAn (Tang et al., 2021), and Spectral (Tran et al., 2018); 2) *input-based*, such as STRIP (Gao et al., 2019) and CD (Huang et al., 2023); 3) *loss-based*, represented by ABL (Li et al., 2021a) and ASSET (Pan et al., 2023). For a fair comparison, we maintain that the number of clean samples per class is 10, extracted from the test dataset. The threshold used in AGPD $\tau_z$ and $\tau_s$ are $e^2$ and 0.05, respectively.

**Datasets and models.** We use CIFAR-10 (Krizhevsky et al., 2009) and Tiny ImageNet (Le & Yang, 2015) as primary datasets to evaluate the detection performance. Additionally, we expand our evaluation to the datasets that are closer to real-world scenarios, such as ImageNet (Deng et al., 2009) subset (200 classes), DTD (Cimpoi et al., 2014), and GTSRB (Houben et al., 2013), of which results are provided in Appendix C.3. Our study employs two model architectures: Preact-ResNet18 (He et al., 2016a) and VGG19-BN (Simonyan & Zisserman, 2014). The results of VGG19-BN are provided in Appendix C.4.1.

**Evaluation metrics.** In this work, the metrics evaluating the performance of backdoor attacks are Accuracy (ACC) and Attack Success Rate (ASR). The metrics used by the defender are True Positive Rate (TPR), False Positive Rate (FPR), and F1 score. In the tables presenting our results, the top performer is highlighted in **bold**, and the runner-up is marked with an underline.

## 5.2 DETECTION EFFECTIVENESS EVALUATION

**All-to-one & all-to-all attacks.**  Tab. 1 showcases the detection performance of AGPD with eight compared methods against 12 backdoor attacks on Preact-ResNet18. For all-to-one attacks and all-to-all attacks, AGPD can achieve averaged TPR of 96.66% on the CIFAR-10 and 99.92% on the Tiny ImageNet, exceeding the runner-up by 18.23% and 11.8% respectively. The averaged FPR of AGPD not only ranks within the top-2 lowest among all detection methods but also approaches a near 0% level. Meanwhile, its average F1 score is 12.71% higher than that of the second-best method.

For the **activation-based** methods, like Beatrix, SCAn and Spectral, we find that they effectively identifies the majority of poisoned samples in attacks where poisoned and clean samples are separated in the activation space. However, its performance deteriorates when this separation is not present, such as CTRL (see t-SNE results in Appendix F). The failure of AC could be caused by the high poisoning ratio. For **input-based** method like STRIP, they exhibit low TPRs in attacks such as WaNet and Input-Aware. This underperformance is likely because the perturbed inputs generated by a poisoned sample also display high entropy in their predictions, similar to those of a clean sample, thereby complicating the distinction between them. CD shows relatively good detection effectiveness across most attacks with an average TPR of 76.76%, although it also suffers from higher FPRs. The reason could be that the masks derived from cognitive distillation for poisoned and clean samples are too similar under $L_1$ norm, leading to misclassification of some clean samples as poisoned. For **loss-based** method like ABL, they perform well in attacks with attacks such as BadNets, Blended, SIG, and CTRL. However, their effectiveness decreases when facing attacks with dynamic triggers, such as WaNet, Input-Aware, and Adap-Blend. These attacks require more training epochs for models to learn the connection between trigger and target label, which means that the loss of poisoned samples does not significantly decrease in the early epochs (Wu et al., 2022). Regarding the ASSET method, we observed potential impacts on detection performance due to differences in the model used compared to the original work. Thus, we provide the results of ASSET on ResNet18 in Appendix E. The evaluation of the model trained on the dataset filtered by AGPD, as well as the detection results under different poisoning ratios on PreActResNet-18, are respectively provided in Appendix C.1 and Appendix C.2.

Table 2: The detection performance of AGPD and the compared methods against multi-target attacks on CIFAR-10. The model structure is Preact-ResNet18. S-T means single trigger, and M-T means multi-trigger.

| Type | Attack | No defense ACC/ASR | AC TPR↑ FPR↓ F1↑ | Beatrix TPR↑ FPR↓ F1↑ | SCAn TPR↑ FPR↓ F1↑ | Spectral TPR↑ FPR↓ F1↑ | STRIP TPR↑ FPR↓ F1↑ | ABL TPR↑ FPR↓ F1↑ | CD TPR↑ FPR↓ F1↑ | ASSET TPR↑ FPR↓ F1↑ | AGPD TPR↑ FPR↓ F1↑ |
|---|---|---|---|---|---|---|---|---|---|---|---|
| S-T | BadNets | 91.38/80.34 | 97.46 0.70 95.64 | 19.38 0.96 30.28 | 0.00 0.00 0.00 | 5.42 16.06 4.34 | 2.78 13.01 2.53 | 1.22 10.98 1.22 | 47.42 31.62 21.95 | 22.84 20.79 14.74 | 98.50 0.08 98.89 |
|  | Blended | 93.57/91.60 | 79.72 0.25 87.61 | 11.76 4.34 15.59 | 0.00 0.00 0.00 | 14.90 15.01 11.92 | 0.18 6.49 0.23 | 2.08 10.88 2.08 | 74.66 55.91 22.03 | 19.64 8.70 19.85 | 98.48 0.01 99.20 |
|  | LF | 93.54/93.82 | 99.72 0.03 99.71 | 20.40 2.33 28.86 | 0.00 0.00 0.00 | 33.78 12.91 27.02 | 5.32 7.88 6.04 | 1.82 10.91 1.82 | 62.02 41.63 23.11 | 0.24 0.66 0.45 | 99.20 0.01 99.55 |
|  | SSBA | 93.28/92.38 | 99.44 0.04 99.52 | 4.50 0.79 8.07 | 0.00 0.00 0.00 | 32.36 13.07 25.89 | 48.26 15.65 33.39 | 11.48 9.84 11.48 | 72.56 25.16 36.37 | 13.52 8.02 14.56 | 98.92 0.02 99.36 |
| M-T | BadNets+Blended+LF+SSBA+SIG | 91.62/92.10 | 58.06 0.08 73.15 | 2.92 2.60 4.62 | 0.00 0.00 0.00 | 14.64 15.05 11.71 | 22.72 13.27 18.77 | 50.10 5.54 50.10 | 63.20 18.60 38.23 | 56.86 54.68 17.52 | 92.02 0.18 95.06 |
|  | Avg. |  | 86.88 0.22 91.13 | 11.79 2.20 17.48 | 0.00 0.00 0.00 | 20.22 14.42 16.18 | 15.85 11.26 12.19 | 13.34 9.63 13.34 | 63.97 34.58 28.34 | 22.62 18.57 13.42 | 97.42 0.06 98.41 |

**Multi-target attacks.**  Tab. 2 summarizes the performance of AGPD and the compared methods in a multi-target attack scenario. In our experiment setting, $\{5, 6, 7, 8, 9\}$ are chosen as the source class and the target labels are set to $t = (y + C) \mod K$, where $C$ equals 5. Single-trigger attack and multi-trigger attack are two categories of multi-target attack. In single-trigger attacks, the same trigger injected into samples from different source classes is classified into their corresponding target classes. In the multi-trigger attack, we use five triggers from different backdoor attacks (*i.e.,* BadNets, Blended, LF, SSBA, and SIG), and each trigger added to the samples in the corresponding class will be classified into its designated target class. We observed that for activation-based methods, the multi-trigger attack poses a greater challenge than single-trigger attacks, whereas loss-based methods seem more robust against multi-trigger attacks. Additionally, the failure of SCAn might be caused by their anomaly detection for target class(es) is not effective in the multi-target attacks. However, compared with baseline methods, AGPD achieves good performance in both single-trigger and multi-trigger attacks, with averages of TPR, FPR, and F1 score at 97.42%, 0.06%, and 98.41%, respectively.

## 5.3 ANALYSIS

**Analysis of CVBT.**  To substantiate the capability of the CVBT metric (*i.e.,* $\rho_{\boldsymbol{x}_0}(\mathcal{D})$ in capturing the characteristics of the circular distribution (*i.e.,* $\mathcal{P}_{\boldsymbol{x}_0}(\mathcal{D})$), here we simulate different circular distributions with varying degrees of dispersion and separation. **(1)** As shown in the left four sub-plots

in Fig. 3, while keeping similar low separation (*i.e.,* one single cluster), the dispersion increases from left to right, *i.e.,* the distribution range increases. Correspondingly, the CVBT score increases. **(2)** As shown in the right four sub-plots in Fig. 3, while keeping similar dispersion (*i.e.,* similar the distribution range), the separation increases from left to right, as two clusters get more distant gradually. Correspondingly, the CVBT score increases. Thus, the claim that *the CVBT score (i.e.,* $\rho_{\boldsymbol{x}_0}(\mathcal{D})$*) is positively proportional to the dispersion and separation of* $\mathcal{P}_{\boldsymbol{x}_0}(\mathcal{D})$ (see Sec. 3.2) is verified. A comparison of the capabilities of CVBT and variance in measuring the characteristics of the circular distribution is also provided in Appendix D.2.

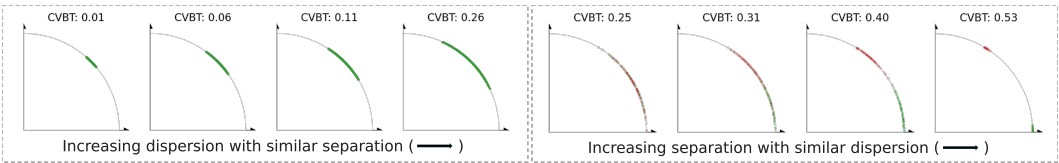

Figure 3: CVBT scores of different GCDs with varying dispersion and separation.

**Statistic of metrics.** In Fig. 4, we present the statistical results of CVBT metric ($\rho$) and its Z-score ($z$) for both all-to-one and all-to-all attacks. In the left of Fig. 4, $\rho$ of the target class are significantly higher than those for clean classes for both attacks. Since clean classes are absent in all-to-all attacks, using $z$ to detect the outliers for identifying the target class could be ineffective. If $z$ doesn't pinpoint the target class, we use a 0.3 threshold as a boundary to identify it, shown as a dashed line in Fig. 4 and validated by the left two images in Fig. 4, where $\rho$ values for target classes surpass this limit in all-to-all attacks. The mid-right image of Fig. 4 displays the distribution of $z$ for the target and clean classes in the all-to-one attacks, clearly separated by the dashed line, which represents the threshold when identifying the target class. Moreover, the right image of Fig. 4 illustrates the mean and standard deviation curves of $z$ of target classes across different convolutional layers of the model. We observe that $z$ tend to be higher in the later intermediate convolutional layers, indicating a stronger separation between poisoned and clean samples at these layers.

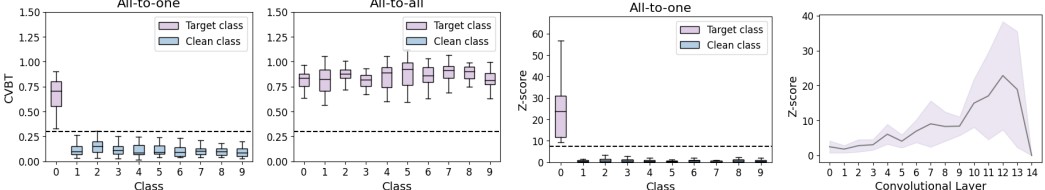

Figure 4: Statistical analysis of $\rho$ and $z$ across classes and convolutional layers using the CIFAR-10 and Preact-ResNet18. **Left:** $\rho$ values for all classes in both all-to-one and all-to-all attacks. **Mid-right:** $z$ for all-to-one attacks. **Right:** Mean and standard deviation of the maximum $z$ across all layers in multiple backdoored models.

**Accuracy of target class identification.** We compare the accuracy of target class identification of AGPD with other three detection methods which are Beatrix (Ma et al., 2023b), SCAn (Tang et al., 2021), and NC (Wang et al.). To evaluate their performance, we trained 120 backdoor models on CIFAR-10. The attack methods contain 8 non-clean label backdoor attacks, where the poisoning ratio ranges from 1% to 10%, and the target label is from 0 to 4. The results of detection accuracy are shown in Fig. 5a. Note that the accuracy of target class identification of AGPD is higher than the compared method under different poisoning ratios.

**Analysis of activation gradient** To illustrate the advantages of activation gradient in sample detection, we also analyze the discriminative characteristics of the activation gradient for the poisoned sample and clean sample in Appendix D.4.

## 5.4 SENSITIVITY TEST

**Influence of the number of clean samples.** In this part, we explore the influence of the size of the additional clean dataset on the detection performance of AGPD. We also consider the scenario that the additional dataset collected by the defender is out of distribution (OOD). We collect the OOD dataset of CIFAR-10 from the same 10 classes of CIFAR-5m (Nakkiran et al., 2020), and we extract 10 samples from each class. The additional clean dataset which is in distribution (ID) is collected

from the test dataset. Fig. 5b shows the results of our method with different sizes of the additional clean dataset. We found that a large number of clean samples can help AGPD decrease FPR close to zero. However, AGPD can still achieve high TPR even in extreme cases, such as one sample per class or even OOD samples. When the number of clean samples in each class is one, the TPR values of AGPD on many attacks are above 90%. In summary, our method necessitates a smaller additional clean dataset.

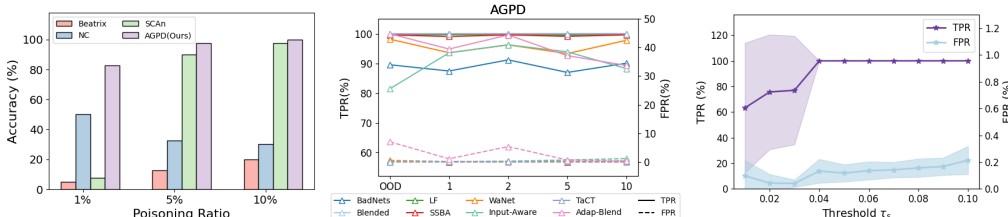

Figure 5: **Left:** Accuracy of AGPD and three compared methods on identifying target class(es). **Middle:** Detection performance of AGPD with varying numbers of clean samples. **Right:** Means and standard deviations of TPR and FPR at different threshold $\tau_s$.

**Influence of threshold $\tau_s$.** In the poisoned sample filtering stage, we aim to eliminate samples scoring below $\tau_s$ at each iteration until no samples remain in the target class. To better understand the impact of $\tau_s$, we design an experiment with varying $\tau_s$ from 0.01 to 0.1. According to Fig. 5c, the TPR of AGPD is relatively low with significant variability at smaller $\tau_s$ values, yet it stabilizes at $100\%$ with increasing $\tau_s$, while the FPR remains consistently low throughout the variation of $\tau_s$. Moreover, it can ensure stable detection performance of AGPD across a broad range of values.

## 5.5 DETECTION EFFECTIVENESS AGAINST ADAPTIVE ATTACKS

**Setup of adaptive attacks.** Here we evaluate AGPD's effectiveness against adaptive attacks, *i.e.,* when the adversary knows its detection strategy. Specifically, the core point in AGPD is the observed characteristic of the gradient circular distribution $\mathcal{P}_{\boldsymbol{x}_0}(\mathcal{D})$, *i.e.,* dispersion and separation of the target GCD (see Sec. 1). Thus, the adaptive adversary aims to break this characteristic. To that end, we design two adaptive attacks. **(1) Adaptive attack 1: Weak clean-label attack for reducing dispersion and separation.** The poisoned samples are constructed based on blending trigger image with target clean images, such that poisoned images are closer to clean images, leading to closer in GCD. This adaptive attack is denoted as **Blended$_\alpha^*$**, being $\alpha$ being the alpha blending coefficient of the trigger image. **(2) Adaptive attack 2: Attacking with inserting noisy samples into the target class for disturbing the target GCD.** We insert some noisy samples into the target class, *i.e.,* randomly picking some clean samples from other classes and changing their labels to target label. As both noisy and poisoned samples are significantly different with target clean samples, the target GCD may vary due to noisy samples. All evaluations are conducted on CIFAR-10 with Preact-ResNet18.

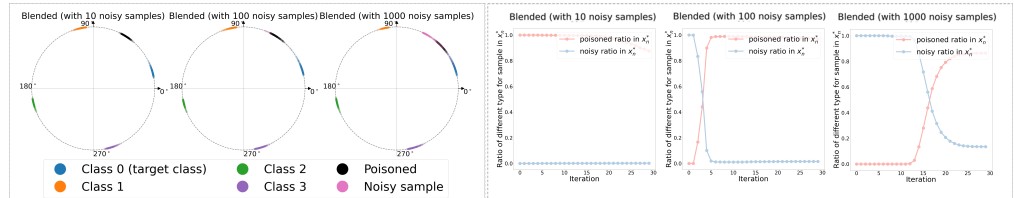

Figure 6: Gradient circular distributions of the target class under Blended and the adaptive Blended$_\alpha^*$ attack.

**Results & Analysis of adaptive attack 1.** We firstly present the GCDs and attack performance (without defense) of Blended$_{\alpha=0.1}^*$, Blended$_{\alpha=0.2}^*$, Blended$_{\alpha=0.3}^*$, respectively, in Fig. 6. It shows that the attack performance is positively proportional to the dispersion and separation of GCD. The detection results of are shown in Tab. 3. When $m = 1$ (*i.e.,* using one single basis vector in $\mathcal{G}_m$), the Z-score is too small to identify the

Table 3: AGPD detection results against Blended$_\alpha^*$ (adaptive attack 1), with different numbers of basis vectors in $\mathcal{G}_m$, *i.e.,* $m = 1/m = 200$.

| Attack | ASR% | Z-score | TPR% | FPR% |
|---|---|---|---|---|
| Blended | 99.75 | 21.34/112.89 | 99.98/99.98 | 0.02/0.02 |
| Blended$_{\alpha=0.1}^*$ | 12.05 | 0.89/2.01 | 0.0/0.0 | 0.01/1.22 |
| Blended$_{\alpha=0.2}^*$ | 51.16 | 2.65/7.06 | 0.0/99.52 | 0.0/10.97 |
| Blended$_{\alpha=0.3}^*$ | 88.33 | 3.42/8.50 | 0.0/99.96 | 0.0/0.86 |

target class, leading to low TPR and low FPR. However, as demonstrated in Sec. 3.2, the basis vector $g(x_{n^*})$ could be extended to a basis set $\mathcal{G}_m = \{g(x_{n_j^*})\}_{j=1,\dots,m}$. When $m = 200$, the Z-scores are much larger. Consequently, even when the attack is weak (*i.e.,* ASR $51\%$ of Blended$^*_{\alpha=0.2}$), AGPD still shows high TPR and low FPR. It demonstrates that increasing the number of basis vectors in $\mathcal{G}_m$ could enhance AGPD's robustness to adaptive weak backdoor attacks.

**Results & Analysis of adaptive attack 2.** As shown in Tab. 4, AGPD still performs very well against the Blended attack with varying noisy samples, and has very high Z-scores. The GCDs of the corresponding poisoned datasets are shown in Fig. 7-Left. It shows that noisy samples may have large angles in GCD, thus the dispersion and separation are still large, leading to high Z-scores. However, when $1,000$ noisy samples exist, the separation degrades, which may affect the sample

Table 4: AGPD detection results against Blended attack with noisy samples (*i.e.,* adaptive attack 2).

| Noisy samples | ASR(%) | Z-score | TPR | FPR |
|---|---|---|---|---|
| 0 | 99.75 | 21.34 | 99.98 | 0.02 |
| 10 | 99.8 | 48.73 | 99.9 | 0.04 |
| 100 | 99.71 | 23.31 | 99.8 | 0.2 |
| 1000 | 99.68 | 35.75 | 100 | 1.8 |

filtering of Stage 3 in AGPD (see Sec. 4.2). Thus, we analyze the trend of the identified far-ending sample $x_{n^*}$ in all iterations. As shown in Fig. 7-Right, we record the proportions of noisy and poisoned samples in all accumulated $x_{n^*}$s. When there are many noisy samples (*e.g.,* 100 or 1,000 noisy samples), noisy samples are identified as $x_{n^*}$ in early iterations, while poisoned samples are gradually identified as $x_{n^*}$ in later iterations. Consequently, both noisy and poisoned samples could be identified as the far-ending basis vector, leading to filtering out of both noisy and poisoned samples. This explains the good performance of AGPD against the adaptive attack with noisy samples.

**In summary**, AGPD shows good performance against above two adaptive attacks, *i.e.,* weak clean-label attack, and attack with noisy samples.

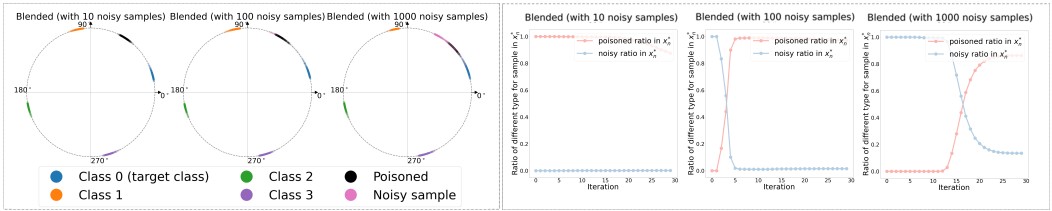

Figure 7: **Left:** The gradient circular distribution of the poisoned dataset with noisy samples of the Blended attack. **Right:** The proportions of noisy or poisoned samples in the accumulated set of far-ending samples $x_{n^*}$ along with the sample filtering iterations.

## 6 CONCLUSION

In this paper, we introduce a novel perspective of gradient circular distribution (GCD). Based on GCD, we observe that the dispersion of GCD of target class is larger, and poisoned samples are separated from clean ones. Inspired by the observation, we propose two practical metrics and design a novel detection method, AGPD. Our experiments demonstrate that this method successfully identifies target class(es) under various backdoor attack scenarios, including all-to-one, all-to-all, and multi-target attacks. Extensive experimental results show that our method achieves good performance on the task of poisoned sample detection. Finally, we believe that the novel perspective of GCD deserves more future explorations, such as its usage in other tasks (*e.g.,* the training-based backdoor defense) and other characteristics.

**Ethics & Reproducibility statements.** This work reveals a common observation of existing backdoor attacks, and provides an advanced backdoor defense method. It will not bring in negative impact to the community. All evaluations are conducted on widely used datasets for image classification, no involvement of ethic issues. Besides, we have provided all important implementation details in Appendix to ensure the reproducibility of all reported results. Our code is based on Backdoorbench(Wu et al., 2022), and we provide a demo of AGPD in the supplementary materials, along with the method of operation, and we promise to release all codes once acceptance.

## ACKNOWLEDGMENTS

This work is supported by the Guangdong Basic and Applied Basic Research Foundation (No. 2024B1515020095), Shenzhen Science and Technology Program (No. RCYX20210609103057050 and JCYJ20240813113608011), Sub-topic of Key R&D Projects of the Ministry of Science and Technology (No. 2023YFC3304804), Longgang District Key Laboratory of Intelligent Digital Economy Security, Guangdong Provincial Special Support Plan - Guangdong Provincial Science and Technology Innovation Young Talents Program (No. 2023TQ07A352), and the National Natural Science Foundation of China No. 62471420.

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

## A    OVERVIEW OF APPENDIX

There are additional materials presented in the Appendix

- Appendix B: Experiment setting details.
    - Appendix B.1: Details of datasets.
    - Appendix B.2: Hyperparameter settings of model training.
    - Appendix B.3: Hyperparameter settings of Backdoor attacks.
- Appendix C:Additional experimental results
    - Appendix C.1:Performance on the model trained by filtered data.
    - Appendix C.2: Different poisoning ratios under PreactResNet-18 on CIFAR-10.
    - Appendix C.3: Evaluations on more datasets.
    - Appendix C.4: Evaluations on more models.
    - Appendix C.5: Comparison to other detection methods.
- Appendix D: Additional analysis of AGPD.
    - Appendix D.1: Details of algorithm and stopping criteria.
    - Appendix D.2: Comparison between CVBT and variance.
    - Appendix D.3: Comparison between cosine distance and radian.
    - Appendix D.4: Analysis of the discriminative degree of activation gradient.
    - Appendix D.5: Computation overhead.
    - Appendix D.6: Stability of AGPD.
- Appendix E: The results of the compared ASSET on ResNet18.
- Appendix F: t-SNE results
- Appendix G: Results for adaptive attacks
- Appendix H: Results for clean-label attacks
- Appendix I: Results for noisy and poisoned samples

## B    EXPERIMENT SETTING DETAILS

### B.1    DATASETS

We evaluate the performance of AGPD on five popular datasets and two model structures. In main paper, we have provide the results of two datasets, including CIFAR-10 (Krizhevsky et al., 2009) and Tiny ImageNet (Le & Yang, 2015). Besides, we extend our evaluations to the datasets which are closer to real-world scenarios, such as ImageNet(subset)-200 (Deng et al., 2009), the Textures dataset DTD (Cimpoi et al., 2014), and the traffic signs dataset GTSRB (Houben et al., 2013). The results of these datasets are provided in Appendix C.3. The details of all datasets are illustrated in Tab. 5 .

Table 5: The information about five datasets.

| Dataset | Categories | Image size | Training samples | Testing samples |
|---------|-----------|-----------|-----------------|----------------|
| CIFAR-10 | 10 | $32 \times 32$ | 50,000 | 10,000 |
| Tiny ImageNet | 200 | $64 \times 64$ | 90,000 | 10,000 |
| ImageNet(subset200) | 200 | $224 \times 224$ | 90,000 | 10,000 |
| GTSRB | 43 | $32 \times 32$ | 39,209 | 12,630 |
| DTD | 47 | $224 \times 224$ | 3,760 | 1,880 |

### B.2    HYPERPARAMETER SETTINGS OF MODEL TRAINING

There are some common training hyperparameters across these attack methods, such as training epoch, learning rate, and optimizer. We display the setting of these common hyperparameters for each datasets in Tab. 6.

Table 6: The common hyperparameters for training across five datasets.

| Dataset | Epoch | Learning rate | Batch size | Optimizer |
|---|---|---|---|---|
| CIFAR-10 | 100 | 0.01 | 128 | SGD |
| Tiny ImageNet | 200 | 0.01 | 128 | SGD |
| ImageNet(subset200) | 200 | 0.1 | 64 | Adam |
| GTSRB | 50 | 0.01 | 128 | SGD |
| DTD | 100 | 0.01 | 64 | SGD |

### B.3 HYPERPARAMETER SETTINGS OF BACKDOOR ATTACKS.

The hyperparameters used in various backdoor attacks are listed in Tab. 7. For illustration purposes, we use CIFAR-10 as an example. If the attack does not have any specific hyper-parameters, we will denote this with '/'. We show the poisoned samples of various backdoor attacks in Fig. 8.

Table 7: The hyper-parameters of implemented backdoor attacks for CIFAR-10.

| | Category | Attack | Parameters | Usage | Value |
|---|---|---|---|---|---|
| All-to-one | non-clean label with sample-agnostic trigger | BadNets | / | / | / |
| | | Blended | $\alpha$ | the transparency of the trigger. | 0.2 |
| | | LF | $\alpha$ | fooling rate | 0.2 |
| | clean label with sample-agnostic trigger | SIG | $\Delta$ | to generate sinusoidal signal. | 40 |
| | | | $f$ | | 6 |
| | clean label with sample-specific trigger | CTRL | $c$ | trigger channel | [2,1] |
| | | | $l$ | trigger location | (12,27) |
| | non-clean label with sample-specific trigger | SSBA | / | / | / |
| | | TaCT | $s-class$ | the trigger will be added to samples in $s-class$ and change their labels to the target label. | a list |
| | | | $c-class$ | samples in $c-class$ will only be added the trigger. | a list |
| | | | $c$ | control the number of samples in $c-class$ | 0.1 |
| | | Adap-Blend | $m$ | the probability of the area being masked. | 0.5 |
| | non-clean label with training control | WaNet | $s$ | warping strength | 0.5 |
| | | | $k$ | grid scale | 4 |
| | | | $\rho_a$ | backdoor probability | =poisoning ratio |
| | | | $\rho_n$ | the noise probability | 0.1 |
| | | Input-Aware | $\lambda_{div}$ | the diversity enforcement regularisation. | 1 |
| | | | $\rho_b$ | the backdoor probability. | =poisoning ratio |
| | | | $\rho_c$ | the cross-trigger probability. | 0.1 |
| All-to-all | non-clean label with sample-agnostic trigger | BadNets-A2A | $K$ | to compute target labels. | 10 |
| | non-clean label with sample-specific trigger | SSBA-A2A | | | |
| Single-trigger Attack | non-clean label with sample-agnostic trigger | BadNets | $C$ | to compute target labels | 5 |
| | | Blended | | | |
| | | LF | | | |
| | non-clean label with sample-specific trigger | SSBA | | | |
| Multi-trigger Attack | non-clean label with sample-agnostic trigger | BadNets+Blended+SSBA+ LF+ SIG | $r$ | the ratio of poisoned samples from each type of trigger | 0.02 |

Table 8: The detection performance of AGPD based on the model, which is trained on the filtered dataset, on CIFAR-10 with Preact-ResNet18.

| dataset | | BadNets | Blended | LF | SSBA | SIG(5%) | WaNet | Input-Aware | TaCT | Adap-Blend |
|---|---|---|---|---|---|---|---|---|---|---|
| Poisoned data | ACC | 91.82 | 93.69 | 93.01 | 92.88 | 93.4 | 89.68 | 91.35 | 93.21 | 92.87 |
| | ASR | 93.79 | 99.75 | 99.05 | 97.06 | 95.43 | 96.94 | 98.17 | 95.95 | 66.17 |
| Filtered data | ACC | 91.9 | 91.52 | 92.02 | 91.7 | 91.91 | 89.98 | 91.14 | 90.71 | 91.25 |
| | ASR | 1.41 | 2.11 | 1.6 | 0.9 | 0.01 | 0.86 | 9.67 | 0.55 | 4.4 |

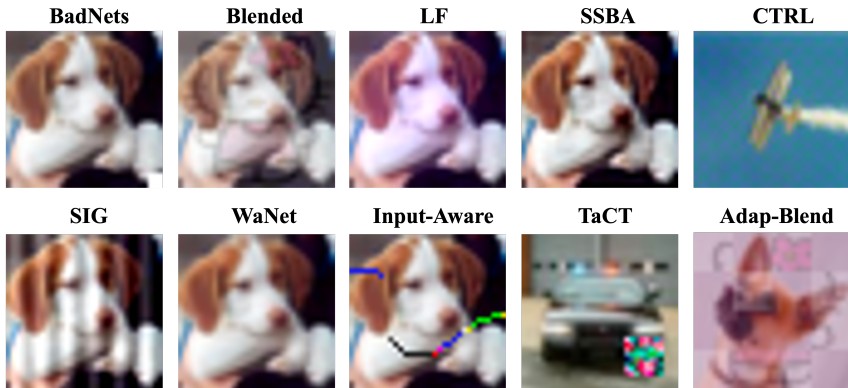

Figure 8: Examples of poisoned samples in various backdoor attacks.

## C ADDITIONAL EXPERIMENTAL RESULTS

### C.1 PERFORMANCE ON THE MODEL TRAINED BY FILTERED DATA

We also use the ACC/ASR of the model trained on the filtered dataset as a metric to evaluate the effectiveness of the model detection. In Tab. 8, we present the ACC/ASR results of the poisoned dataset under different attacks after being filtered and trained using the AGPD method on CIFAR-10 and PreActResNet-18. Our preliminary evaluations on CIFAR-10 and PreAct-ResNet-18 demonstrate that training on the dataset filtered by AGPD can achieve high ACC and low ASR. The AGPD method thus ensures the model's performance while resisting backdoor attacks.

### C.2 DIFFERENT POISONING RATIOS UNDER PREACTRESNET-18 ON CIFAR-10

We evaluated the detection effectiveness of AGPD and four other methods, chosen from activation-based, input-based, and loss-based detection methods, under varying poisoning ratios. There are four poisoning ratios used: $\{0.5\%, 1\%, 5\%, 10\%\}$, covering a range from low to high poisoning ratios. As illustrated in Fig. 9, the performance of most detectors is notably influenced by the poisoning ratio. Particularly, a low poisoning ratio (*e.g.,* 0.5%) presents a substantial challenge for most detectors, with the TPRs of AC and SCAn almost nearing zero. However, our method can achieve good performance in this situation, with TPR around 90% and FPR lower than that of other detectors under most attacks. And with the poisoning ratio increased, the performance of our method is still stable.

### C.3 EVALUATIONS ON MORE DATASETS

In this section, we present the results of AGPD and the compared methods on these datasets, including DTD (Cimpoi et al., 2014), GTSRB (Houben et al., 2013), and ImageNet subset (200 classes) (Deng et al., 2009). The datasets can be categorized into two types: balanced datasets (*e.g.,* DTD and ImageNet subset (200 classes)) and imbalanced datasets (*e.g.,* GTSRB). In balanced datasets, each category has the same number of samples. For example, DTD has 376 samples per class, and ImageNet-200 has 500 samples per class. In contrast, the number of samples per category in GTSRB varies from 210 to 2,250. Besides, We use Preact-ResNet18 (He et al., 2016a) as the model

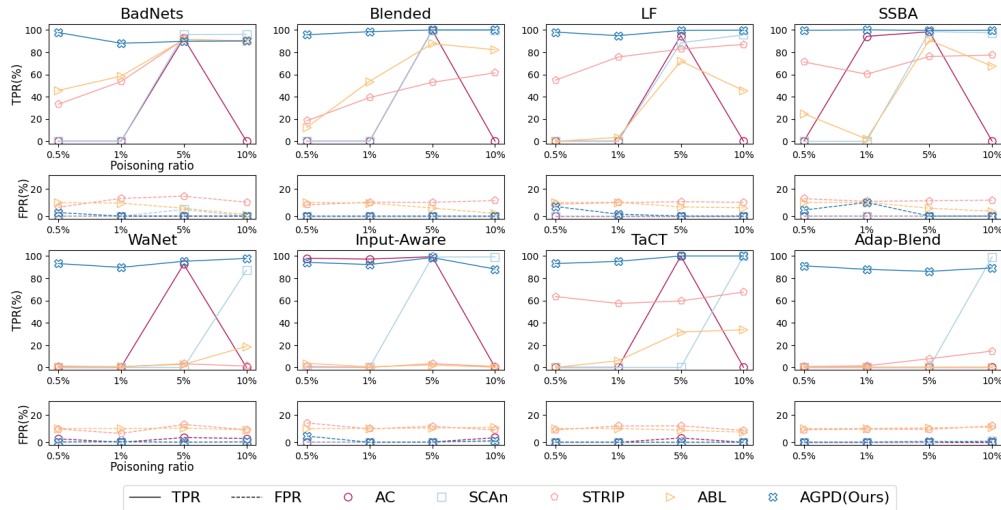

Figure 9: Detection performance of AGPD and the compared detectors with poisoing ratios ranging from 0.5% to 10%.

architecture when training on DTD and GTSRB, and adopt ResNet50 (He et al., 2016b) for ImageNet subset (200 classes). Specifically, we compare AGPD with four detection methods: activation-based method (*e.g.,* SCAn (Tang et al., 2021)), input-based method (*e.g.,* STRIP (Gao et al., 2019)), and loss-based methods (*e.g.,* ABL (Li et al., 2021a) and ASSET (Pan et al., 2023)) on DTD and GTSRB. The results are shown in Tab. 9. Besides, the results of AGPD on ImageNetsubset (200 classes), are displayed in Tab. 10.

Table 9: The detection performance of AGPD and compared detectors on DTD and GTSRB. The results are evaluated on Preact-ResNet18.

| Dataset | Attack | No defense ACC/ASR | SCAn TPR↑ | FPR↓ | F1↑ | STRIP TPR↑ | FPR↓ | F1↑ | ABL TPR↑ | FPR↓ | F1↑ | ASSET TPR↑ | FPR↓ | F1↑ | AGPD TPR↑ | FPR↓ | F1↑ |
|---|---|---|---|---|---|---|---|---|---|---|---|---|---|---|---|---|---|
| DTD | BadNets | 51.97/98.32 | 90.43 | **0.21** | **94.05** | 83.78 | 12.71 | 56.20 | 76.33 | 2.63 | 76.43 | 96.01 | 8.16 | 71.15 | **99.47** | 1.60 | 93.03 |
| | Blended | 51.86/94.62 | 82.18 | **0.30** | 88.76 | 95.74 | 13.21 | 60.74 | 88.83 | 1.24 | 88.95 | 0.80 | 3.04 | 1.25 | **100.0** | 1.89 | 92.27 |
| | WaNet | 42.71/26.41 | 86.08 | **1.06** | 88.01 | 77.84 | 14.03 | 51.14 | 0.28 | 11.00 | 0.27 | 1.70 | 2.33 | 2.61 | **100.0** | 1.88 | 92.27 |
| | Input-Aware | 45.85/85.54 | 0.00 | **0.00** | 0.00 | 13.07 | 12.91 | 11.38 | 20.45 | 8.92 | 20.19 | 0.00 | 0.72 | 0.00 | **98.58** | 0.59 | 96.73 |
| | Adap-Blend | 49.41/85.65 | 77.13 | **0.92** | 83.21 | 32.18 | 16.19 | 23.01 | 6.91 | 10.34 | 6.67 | 4.26 | 2.99 | 6.49 | **100.0** | 1.18 | 95.07 |
| | Avg. | | 67.16 | 0.50 | 70.81 | 60.52 | 13.81 | 40.50 | 38.56 | 6.83 | 38.50 | 20.55 | 3.45 | 16.30 | **99.61** | 1.43 | 93.88 |
| GTSRB | BadNets | 96.35/95.02 | 94.62 | 4.00 | 82.06 | 95.54 | 15.39 | 57.20 | 73.71 | 2.92 | 73.71 | **100.0** | 47.79 | 31.74 | **100.0** | **0.27** | 98.80 |
| | Blended | 98.17/100.0 | 83.47 | 7.54 | 66.42 | **100.0** | 11.58 | 65.74 | 80.54 | 2.16 | 80.55 | 99.36 | 41.33 | 34.77 | 95.87 | **0.30** | 96.57 |
| | WaNet | 97.05/96.16 | 60.83 | **0.00** | 75.63 | 7.83 | 14.34 | 6.59 | 0.00 | 11.03 | 0.00 | 44.07 | 64.79 | 12.12 | **100.0** | 0.20 | 99.12 |
| | Input-Aware | 97.91/95.64 | 52.15 | **0.00** | 68.54 | 1.99 | 10.44 | 2.03 | 0.00 | 11.03 | 0.00 | 3.07 | 59.18 | 0.96 | **79.60** | 0.00 | 88.64 |
| | Adap-Blend | 97.66/80.42 | 50.74 | 3.95 | 54.48 | 20.48 | 12.40 | 17.63 | 0.00 | 11.11 | 0.00 | 81.10 | 22.48 | 42.30 | **93.44** | 0.23 | 95.58 |
| | Avg. | | 68.36 | 3.10 | 69.43 | 45.17 | 12.83 | 29.84 | 30.85 | 7.65 | 30.85 | 65.52 | 47.12 | 24.38 | **93.78** | **0.20** | 95.74 |

Table 10: The detection performance of AGPD on ImageNet-200. The results are evaluated on Preact-ResNet18.

| Dataset | Attack | No defense ACC/ASR | AGPD TPR↑ | FPR↓ | F1↑ |
|---|---|---|---|---|---|
| ImageNet-200 | BadNet | 78.57/80.03 | 94.62 | 0.00 | 97.24 |
| | Blended | 79.95/99.93 | 100.0 | 0.47 | 97.93 |
| | Adap-Blend | 72.3/93.17 | 99.98 | 0.59 | 99.19 |
| | Avg. | | 98.20 | 0.35 | 98.12 |

As shown in Tables 9 and 10, AGPD achieves high performance on these datasets, with average TPRs of 99.61%, 93.78%, and 98.20%, respectively. Meanwhile, the average FPRs are 1.43%, 0.2%, and 0.35%. The experimental results demonstrate that our method is adaptable not only to balanced and imbalanced datasets but also to datasets with various image sizes.

Regarding the compared detection methods, we draw conclusions similar to those reported in our main paper. Specifically, SCAn fails when the distinction between poisoned and clean samples in the activation space is not obvious. STRIP struggles to effectively identify poisoned samples in the training dataset under attacks such as WaNet and Input-Aware. Similarly, ABL encounters challenges in achieving satisfactory detection performance against complex triggers, including those used in WaNet, Input-Aware, and Adap-Blend attacks.

## C.4 EVALUATIONS ON MORE MODELS

### C.4.1 EVALUATIONS ON VGG19-BN

**All-to-one & all-to-all attacks.** The results of VGG19-BN under all-to-one and all-to-all attacks are shown in Tab. 11. It can be seen that even changing the model architecture, the detection performance of our method is still stable, achieving 96.46% TPR on CIFAR-10 and 97.75% on Tiny ImageNet, higher than that of the second-best 16.55% and 7.89%, respectively. Besides, the F1 score of AGPD are 90.90% on CIFAR-10 and 98.36% on Tiny ImageNet, exceeding the runner-up by 33.59% and 16.27%, respectively. The results indicate the dispersion of the activation gradients of poisoned samples and clean samples could exist across model structures, demonstrating the robust adapt ability of our method.

For the compared methods, we found that the model architecture significantly influences activation-based methods. For instance, AC can identify a small portion of poisoned samples in all-to-one attacks with a 10% poisoning ratio. Beatrix performs better on VGG19-BN, with the average TPR 20.78% higher and the F1 score 17.39% greater than on Preact-ResNet18. Conversely, SCAn struggles to detect a large number of poisoned samples in WaNet and Input-Aware attacks on the VGG19-BN.

Table 11: The detection performance of AGPD and compared detectors on CIFAR-10 and Tiny ImageNet. The results are evaluated on VGG19-BN.

| Dataset | Attack | No defense ACC/ASR | AC TPR↑ | AC FPR↓ | AC F1↑ | Beatrix TPR↑ | Beatrix FPR↓ | Beatrix F1↑ | SCAn TPR↑ | SCAn FPR↓ | SCAn F1↑ | Spectral TPR↑ | Spectral FPR↓ | Spectral F1↑ | STRIP TPR↑ | STRIP FPR↓ | STRIP F1↑ | ABL TPR↑ | ABL FPR↓ | ABL F1↑ | CD TPR↑ | CD FPR↓ | CD F1↑ | ASSET TPR↑ | ASSET FPR↓ | ASSET F1↑ | AGPD TPR↑ | AGPD FPR↓ | AGPD F1↑ |
|---|---|---|---|---|---|---|---|---|---|---|---|---|---|---|---|---|---|---|---|---|---|---|---|---|---|---|---|---|---|
| CIFAR-10 | BadNets | 91.82/93.79 | 0.00 | 6.42 | 0.00 | 52.90 | 3.44 | 57.55 | 68.90 | 0.35 | 80.08 | 28.28 | 0.02 | 44.02 | 82.88 | 10.84 | 59.10 | 76.68 | 2.59 | 76.68 | 72.98 | 13.46 | 49.62 | 4.54 | 2.33 | 7.24 | 99.72 | 6.90 | 76.16 |
| | Blended | 93.69/99.75 | 0.00 | 14.79 | 0.00 | 99.22 | 9.97 | 68.68 | 96.72 | 0.00 | 98.33 | 28.50 | 0.00 | 44.36 | 47.06 | 10.74 | 38.61 | 78.08 | 2.44 | 78.08 | 99.94 | 99.87 | 18.19 | 0.20 | 52.35 | 0.07 | 99.96 | 0.02 | 99.90 |
| | LF | 93.01/99.05 | 0.00 | 6.76 | 0.00 | 99.46 | 6.40 | 77.39 | 83.22 | 0.00 | 90.83 | 28.50 | 0.00 | 44.36 | 89.80 | 8.28 | 67.94 | 40.80 | 6.58 | 40.80 | 0.10 | 0.03 | 0.20 | 1.74 | 0.97 | 3.15 | 99.50 | 0.03 | 99.62 |
| | SSBA | 92.88/97.06 | 4.74 | 14.08 | 4.10 | 0.10 | 6.19 | 0.13 | 99.76 | 0.80 | 91.15 | 2.18 | 2.92 | 3.39 | 72.38 | 12.33 | 51.08 | 25.16 | 8.32 | 25.16 | 84.78 | 60.67 | 23.20 | 0.00 | 0.00 | 0.00 | 98.94 | 4.81 | 81.69 |
| | SIG | 93.40/95.43 | 0.00 | 13.84 | 0.00 | 100.0 | 3.56 | 74.70 | 94.80 | 0.00 | 97.33 | 30.00 | 0.00 | 46.15 | 98.96 | 6.20 | 62.47 | 94.84 | 5.53 | 63.23 | 88.24 | 48.16 | 28.39 | 100.0 | 47.35 | 18.19 | 99.60 | 0.02 | 99.62 |
| | CTRL | 95.52/98.8 | 15.52 | 17.08 | 7.06 | 22.20 | 4.79 | 20.83 | 5.80 | 0.00 | 10.96 | 23.20 | 14.57 | 11.60 | 94.36 | 10.19 | 48.64 | 86.72 | 5.96 | 57.81 | 88.16 | 23.29 | 44.33 | 100.0 | 5.81 | 64.42 | 90.84 | 2.06 | 78.99 |
| | WaNet | 89.68/96.94 | 0.00 | 20.48 | 0.00 | 2.45 | 9.64 | 2.51 | 0.00 | 0.00 | 0.00 | 29.03 | 0.06 | 44.81 | 1.92 | 12.02 | 1.76 | 13.18 | 9.67 | 12.76 | 98.85 | 98.54 | 18.21 | 0.00 | 38.57 | 0.00 | 85.35 | 0.09 | 91.67 |
| | Input-Aware | 90.82/98.17 | 10.52 | 7.01 | 11.80 | 2.03 | 5.63 | 2.59 | 65.04 | 0.03 | 78.69 | 1.88 | 2.86 | 2.90 | 0.98 | 8.60 | 1.07 | 14.83 | 9.50 | 14.35 | 84.26 | 58.53 | 23.70 | 0.06 | 10.01 | 0.07 | 92.96 | 0.06 | 96.09 |
| | TaCT | 93.21/95.95 | 0.00 | 6.46 | 0.00 | 97.24 | 6.79 | 75.29 | 99.22 | 0.00 | 99.61 | 30.00 | 0.00 | 46.15 | 75.30 | 14.19 | 49.70 | 26.94 | 8.12 | 26.94 | 83.42 | 74.00 | 19.64 | 0.42 | 0.66 | 0.79 | 100.0 | 0.25 | 98.90 |
| | Adap-Blend | 92.87/66.17 | 4.68 | 16.07 | 3.75 | 0.22 | 7.21 | 0.27 | 29.68 | 1.80 | 40.68 | 1.36 | 3.03 | 2.12 | 7.34 | 11.31 | 7.02 | 0.02 | 11.11 | 0.02 | 100.0 | 100.0 | 18.18 | 0.00 | 0.01 | 0.00 | 99.92 | 8.17 | 73.07 |
| | BadNets-A2A | 91.93/74.40 | 79.46 | 0.02 | 88.49 | 28.42 | 6.66 | 30.17 | 0.00 | 0.00 | 0.00 | 0.00 | 1.67 | 0.00 | 1.54 | 11.72 | 1.49 | 4.86 | 10.57 | 4.86 | 58.16 | 20.01 | 34.39 | 5.86 | 3.67 | 8.44 | 95.92 | 0.02 | 97.83 |
| | SSBA-A2A | 93.46/87.84 | 99.36 | 0.93 | 95.66 | 31.44 | 7.17 | 32.08 | 0.00 | 0.00 | 0.00 | 0.00 | 1.67 | 0.00 | 19.98 | 16.14 | 15.07 | 1.18 | 10.98 | 1.18 | 99.98 | 99.96 | 18.18 | 0.24 | 0.11 | 0.47 | 94.82 | 0.02 | 97.27 |
| | Avg. | | 17.86 | 10.33 | 17.57 | 44.64 | 6.45 | 36.85 | 52.76 | 0.25 | 57.31 | 16.91 | 2.23 | 24.15 | 49.38 | 13.05 | 33.66 | 38.61 | 7.61 | 33.49 | 79.91 | 58.04 | 24.69 | 17.76 | 13.49 | 8.57 | 96.46 | 1.87 | 90.90 |
| Tiny ImageNet | BadNets | 56.12/99.90 | 0.10 | 20.08 | 0.07 | 97.73 | 9.37 | 69.30 | 99.88 | 0.00 | 99.94 | 15.67 | 0.00 | 27.09 | 99.99 | 11.24 | 66.41 | 96.66 | 0.37 | 96.66 | 96.43 | 9.16 | 69.15 | 100.0 | 48.50 | 31.42 | 99.67 | 0.12 | 99.30 |
| | Blended | 55.53/97.57 | 0.00 | 7.47 | 0.00 | 92.85 | 4.40 | 79.90 | 78.26 | 0.00 | 87.80 | 15.67 | 0.00 | 27.09 | 95.55 | 13.88 | 59.63 | 96.78 | 0.36 | 96.78 | 82.38 | 34.19 | 33.62 | 1.32 | 4.65 | 1.84 | 99.99 | 0.03 | 99.85 |
| | LF | 55.21/98.51 | 0.00 | 5.74 | 0.00 | 22.73 | 2.02 | 32.26 | 52.25 | 0.00 | 68.64 | 15.67 | 0.00 | 27.09 | 21.09 | 11.82 | 18.55 | 48.83 | 5.69 | 48.83 | 96.43 | 10.55 | 66.18 | 99.08 | 17.03 | 56.24 | 99.92 | 0.03 | 99.84 |
| | SSBA | 55.97/97.69 | 0.00 | 8.88 | 0.00 | 42.53 | 3.96 | 47.73 | 78.05 | 0.00 | 87.67 | 15.67 | 0.00 | 27.09 | 99.92 | 11.77 | 65.33 | 89.65 | 1.15 | 89.65 | 93.11 | 19.00 | 51.14 | 100.0 | 47.66 | 31.80 | 99.94 | 0.02 | 99.86 |
| | WaNet | 58.33/90.35 | 0.02 | 5.62 | 0.03 | 0.00 | 1.51 | 0.00 | 99.94 | 0.00 | 99.96 | 13.93 | 0.19 | 24.08 | 17.65 | 11.57 | 15.38 | 83.54 | 2.39 | 80.85 | 99.09 | 0.30 | 98.19 | 100.0 | 0.73 | 97.28 | 99.97 | 0.12 | 99.40 |
| | Input-Aware | 57.5/99.75 | 0.05 | 5.64 | 0.07 | 0.53 | 2.90 | 0.83 | 98.11 | 0.00 | 99.05 | 15.72 | 0.00 | 27.17 | 15.49 | 12.35 | 13.19 | 64.97 | 4.31 | 62.87 | 95.88 | 8.36 | 70.73 | 97.18 | 15.79 | 58.19 | 82.54 | 0.00 | 90.43 |
| | TaCT | 54.93/91.25 | 0.00 | 8.16 | 0.00 | 54.87 | 2.62 | 61.50 | 44.87 | 0.00 | 61.95 | 15.05 | 0.08 | 26.00 | 99.91 | 17.27 | 56.24 | 56.75 | 4.81 | 56.75 | 74.04 | 42.16 | 26.75 | 99.35 | 51.83 | 31.05 | 99.99 | 0.24 | 98.94 |
| | Adap-Blend | 54.55/96.35 | 25.95 | 21.18 | 16.39 | 0.68 | 10.67 | 0.69 | 34.92 | 0.01 | 51.73 | 14.93 | 0.08 | 25.81 | 67.20 | 15.69 | 43.58 | 1.09 | 10.99 | 1.09 | 81.50 | 23.87 | 41.12 | 100.0 | 57.36 | 30.35 | 99.99 | 0.16 | 99.27 |
| | Avg. | | 3.27 | 10.35 | 2.07 | 38.99 | 4.68 | 36.53 | 73.28 | 0.00 | 82.09 | 15.29 | 0.04 | 26.43 | 64.60 | 13.20 | 42.29 | 67.28 | 3.76 | 66.69 | 89.86 | 18.45 | 57.11 | 87.12 | 30.44 | 42.27 | 97.75 | 0.09 | 98.36 |

**Performance of AGPD with VGG19-BN under various poisoning ratios.** We estimate the detection performance of AGPD against various backdoor attacks with different poisoning ratios and compare our method with four detectors. The results are displayed in Fig. 10. It can be seen that our method achieves a higher TPR under most attacks compared to other methods, which also maintain relatively low FPR.

## C.5 COMPARISON TO OTHER DETCTION METHODS

In this section, we compare our method alongside CT (Qi et al., 2023b) and IBD-PSC (Hou et al., 2024). To ensure a fair comparison, we maintained experimental settings of attacks (*i.e.,*, learning rate and training epoch $E$) consistent with our main experiment, as introduced in Sec 5.1. The details are shown as follows:

- **Dataset and model architecture:** We compare AGPD with CT and IBD-PSC on CIFAR-10 dataset and PreAct-ResNet18.

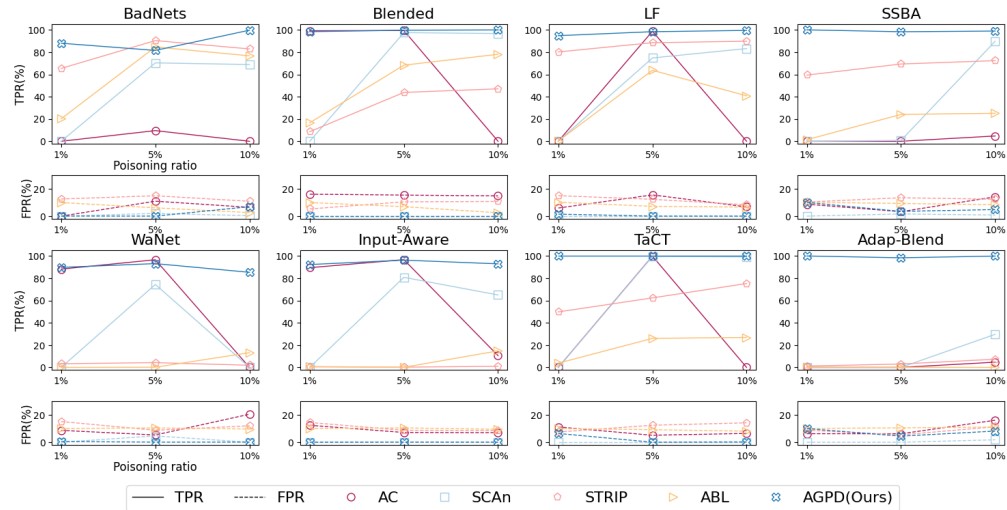

Figure 10: Detection performance of AGPD and the compared detectors with poisoning ratios ranging from 1% to 10%.

- **Attack method and poisoning ratio:** We choose four classical attack methods which are BadNets, Blended, SSBA, and WaNet. The poisoning ratio is 10%.

- **Hyper-parameters for CT:** We set the number of clean samples is 2,000 and confusion factor $\lambda$ is 20. Besides, we set confusion iteration $K = 6$, confusion iter $m = 6000$, a learning rate of 0.001, and the distillation ratios $\{r_1 = \frac{1}{2}, r_2 = \frac{1}{5}, r_3 = \frac{1}{25}, r_4 = \frac{1}{50}, r_5 = \frac{1}{100}\}$.

- **Hyper-parameters of IBD-PSC:** The number of clean samples is 2,000. The hyper-parameters for error rate, threshold $T$, and amplifying coefficient for the selected BN layer set to 0.6, 1.5, and 0.9, respectively.

Table 12: The detection performance of AGPD and other detection methods on CIFAR-10. The results are evaluated on Preact-ResNet18.

| Attack | CT | | | IBD-PSC | | | AGPD | | |
|---|---|---|---|---|---|---|---|---|---|
| | TPR↑ | FPR↓ | F1↑ | TPR↑ | FPR↓ | F1↑ | TPR↑ | FPR↓ | F1↑ |
| Badnet | 95.94 | **0.01** | **97.90** | **99.82** | 11.02 | 64.14 | 90.06 | 0.03 | 94.65 |
| Blended | 99.96 | 1.36 | 94.21 | 99.47 | 11.42 | 63.17 | **99.98** | **0.02** | **99.88** |
| SSBA | **100.00** | 0.08 | **99.63** | 85.64 | 10.86 | 60.07 | 99.62 | **0.04** | **99.63** |
| WaNet | 91.64 | **0.17** | 94.89 | 96.24 | 11.41 | 61.78 | **97.80** | 0.31 | **97.40** |

The experimental results are shown in Tab. 12. It can be seen that both our method and the compared methods achieve high performance in detecting poisoned samples. Notably, our method performs well in minimizing misclassification of clean samples during the detection task, resulting in a lower FPR.

# D ADDITIONAL ANALYSIS OF AGPD

## D.1 DETAILS OF ALGORITHM AND STOPPING CRITERIA

### D.1.1 THE DESCRIPTION OF ALGORITHM

The statement of the algorithm we used in Stage3 is described in Algorithm 1. An example of JS divergence across iterations is provided in Fig. 11. Assuming ground truth is known for samples in the target class, we can obtain True Positives (TP) and False Positives (FP) for each iteration. It can be observed that an optimal iteration exists where JS divergence is minimal and stabilizes. The

rationale behind the trends in JS divergence is that in the early stages of filtering, the far-end basis is primarily updated by genuinely poisoned samples, effectively guiding the identification of such samples. As the process progresses into the middle stages, most poisoned samples have been filtered out, and the influence of the far-end basis on the remaining clean samples becomes minimal, resulting in little change in trust scores and small JS divergence. However, as the process extends into later stages, and more clean samples are inevitably filtered, the far-end basis is updated by these clean samples, leading to significant changes in the distribution of trustworthiness scores and an increase in JS divergence.

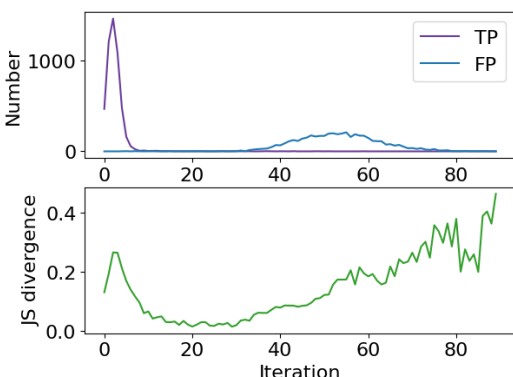

Figure 11: Trends of TP, FP, and JS divergence according to the iteration $t$.

---

**Algorithm 1** Filtering out poisoned samples within the identified target class(es).

---

**Input:** The identified target class $k^*$, the subset $\mathcal{D}_{bd}^{k^*}$, selected layer $l^*$, the reference $(\boldsymbol{x}_0, k^*)$, and filtering threshold $\tau_s$.
**Output:** Suspected set $\mathcal{D}_{sus}^{k^*}$ and purified set $\mathcal{D}_{bd}^{k^*} \backslash \mathcal{D}_{sus}^{k^*}$.
 1: Compute the GCDs of the set $\mathcal{D}_{bd}^{k^*}$, referred to $\{\theta_{\boldsymbol{x}_0}(\boldsymbol{x}_i)\}_{i=1}^{n_{k^*}}$, corresponding to the reference $(x_0, k^*)$, , according to Eq.(2).
 2: Find the farthest activation gradient $g(\boldsymbol{x}_{n^*})$ according to $n^* = \arg\max_{i \in \{1, \dots, n_{k^*}\}} \theta_{\boldsymbol{x}_0}(\boldsymbol{x}_i)$.
 3: Set $\mathcal{D}_{sus}^{k^*} = \emptyset$, $JS = \{\}$, and iteration $t = 0$.
 4: **while** $\mathcal{D}_{bd}^{k^*} \backslash \mathcal{D}_{sus}^{k^*} \neq \emptyset$ **do**
 5: $\quad$ Calculate the distribution of $\{s_{\boldsymbol{x}_0}(\boldsymbol{x}_i)\}_{\boldsymbol{x}_i \in \mathcal{D}_{bd}^{k^*}}$ according to Eq.(4).
 6: $\quad$ Add samples $(\boldsymbol{x}_i, k^*)$ whose $s_{\boldsymbol{x}_0}(\boldsymbol{x}_i)$ is smaller than $\tau_s$ to $\mathcal{D}_{sus}^{k^*}$, and remove them from $\mathcal{D}_{bd}^{k^*}$.
 7: $\quad$ **if** $t > 0$ **then**
 8: $\quad\quad$ Calculate the JS divergence between the distribution of $\{s_{\boldsymbol{x}_0}(\boldsymbol{x}_i)\}_{\boldsymbol{x}_i \in \mathcal{D}_{bd}^{k^*}}$ in the iteration $t$ and $t-1$.
 9: $\quad\quad$ Add the JS divergence to $JS$.
10: $\quad$ **end if**
11: $\quad$ $t = t + 1$.
12: **end while**
13: Find an appropriate iteration $t^*$ according to the stopping criteria.
14: Remain samples $(\boldsymbol{x}_i, y)$ filtered out before the iteration $t^*$ in $\mathcal{D}_{sus}^{k^*}$.

---

### D.1.2 THE STOPPING CRITERIA

To find an appropriate stopping iteration $t^*$, we utilize the sliding window method to analyze the changes in JS divergence across all iterations. Our goal is to identify the iteration $t$ where the JS divergence is minimal and stabilizes. Let the width of the window be $w$, and let the JS divergence at each iteration $t$ be $JS(t)$, where $t \in \{0, \dots, T-1\}$. The average $\mu_m$ and standard deviation $\sigma_m$ of each window starting at position $m$ are defined as follows:

$$\begin{cases} \mu_m = \dfrac{1}{w} \sum_{j=0}^{w-1} JS(m+j), \\[2ex] \sigma_m = \sqrt{\dfrac{1}{w} \sum_{j=0}^{w-1} \left(JS(m+j) - \mu_m\right)^2}. \end{cases} \tag{6}$$

Each window can be represented by a score $S_m$ that combines the value of the average with the standard deviation. Since we are mainly focusing on stabilization, we design a metric as described in Eq. 7, which amplifies the contribution of the standard deviation.

$$S_m = \mu_m + \beta \sigma_m. \tag{7}$$

After computing the score of all windows, we choose the window starting at $m$ with the minimum score:

$$m^* = \arg\min_m S_m. \tag{8}$$

The stopping iteration is the middle of the optimal window, which is $t^* = m^* + \frac{w}{2}$. In our method, we adopt $w = 10$ and $\beta = 5$.

### D.1.3 ANALYSIS OF THE HYPER-PARAMETERS IN THE STOPPING CRITERIA

In this section, we tested how changes in window size and beta affect detection performance across different attacks on Tab. 13 and Tab. 14. Our findings show that within a certain range, both parameters maintain strong detection performance, making our approach reliable under various conditions. A small window size might cause the detection process to stop too early due to fluctuations when a few poisoned samples are detected. A large window size might include clean samples, increasing the mean JS divergence and causing an early stop. For $\beta$, a small value might focus on the later stages with high variance, while a large beta might increase the false positive rate by selecting stable stages. Balancing these parameters is crucial for optimal detection.

Table 13: The detection performance of AGPD with different window sizes on CIFAR-10. The results are evaluated on Preact-ResNet18.

| Window size→ | 2 | | | 4 | | | 6 | | | 8 | | | 10 | | | 12 | | | 14 | | | 16 | | | 18 | | | 20 | | |
|---|---|---|---|---|---|---|---|---|---|---|---|---|---|---|---|---|---|---|---|---|---|---|---|---|---|---|---|---|---|---|
| Attack↓ | TPR↑ | FPR↓ | F1↑ | TPR↑ | FPR↓ | F1↑ | TPR↑ | FPR↓ | F1↑ | TPR↑ | FPR↓ | F1↑ | TPR↑ | FPR↓ | F1↑ | TPR↑ | FPR↓ | F1↑ | TPR↑ | FPR↓ | F1↑ | TPR↑ | FPR↓ | F1↑ | TPR↑ | FPR↓ | F1↑ | TPR↑ | FPR↓ | F1↑ |
| BadNets | 98.3 | 0.96 | 94.99 | 90.06 | 0.03 | 94.65 | 90.06 | 0.03 | 94.65 | 90.06 | 0.03 | 94.65 | 90.06 | 0.03 | 94.65 | 90.06 | 0.03 | 94.65 | 90.06 | 0.03 | 94.65 | 90.06 | 0.03 | 94.65 | 90.06 | 0.03 | 94.65 | 85.36 | 0 | 92.08 |
| Blended | 93.08 | 2.11 | 88.71 | 93.36 | 0.77 | 93.24 | 94.74 | 1.1 | 92.65 | 91.7 | 0.14 | 94.85 | 94.65 | 0.175 | 96.385 | 90.89 | 0.13 | 94.465 | 93.8 | 0.315 | 95.31 | 93.33 | 0.305 | 95.125 | 91.45 | 0.145 | 94.725 | 90.89 | 0.13 | 94.465 |
| LF | 99.32 | 0 | 99.64 | 99.28 | 0 | 99.62 | 99.18 | 0 | 99.58 | 99.04 | 0 | 99.51 | 99.8 | 0.07 | 99.6 | 99.8 | 0.07 | 99.6 | 99.8 | 0.07 | 99.6 | 99.8 | 0.07 | 99.6 | 98.24 | 0 | 99.1 | 97.86 | 0 | 98.91 |
| SIG | 99.76 | 3.17 | 76.75 | 100 | 0.04 | 99.66 | 100 | 0.04 | 99.66 | 100 | 0.04 | 99.66 | 100 | 0.04 | 99.66 | 98.84 | 3.83 | 72.78 | 98.84 | 3.99 | 72 | 98.84 | 3.57 | 74.12 | 98.2 | 3.17 | 76.02 | 98.84 | 3.17 | 76.32 |
| SSBA | 99.16 | 0.02 | 99.5 | 99.04 | 0.02 | 99.45 | 99.04 | 0.02 | 99.45 | 98.88 | 0.01 | 99.39 | 99.62 | 0.04 | 99.63 | 98.88 | 0.01 | 99.39 | 99.62 | 0.04 | 99.63 | 98.34 | 0.01 | 99.13 | 97.9 | 0.01 | 98.91 | 97.9 | 0.01 | 98.91 |
| TACT | 99.9 | 0.01 | 99.89 | 99.8 | 0.01 | 99.85 | 99.88 | 0.01 | 99.88 | 99.88 | 0.01 | 99.88 | 100 | 0.07 | 99.68 | 100 | 0.07 | 99.68 | 100 | 0.07 | 99.68 | 99.18 | 0 | 99.57 | 99.18 | 0 | 99.57 | 99.18 | 0 | 99.57 |
| WANet | 95.18 | 0.09 | 97.07 | 95.18 | 0.09 | 97.07 | 94.43 | 0.05 | 96.89 | 93.58 | 0.04 | 96.5 | 97.8 | 0.31 | 97.4 | 91.87 | 0.02 | 95.68 | 91.02 | 0.02 | 95.21 | 91.87 | 0.02 | 95.68 | 89.55 | 0.02 | 94.41 | 88.01 | 0.01 | 93.58 |

Table 14: The detection performance of AGPD with different $\beta$ on CIFAR-10. The results are evaluated on Preact-ResNet18.

| $\beta \rightarrow$ | 1 | | | 3 | | | 5 | | | 7 | | | 9 | | | 11 | | | 13 | | |
|---|---|---|---|---|---|---|---|---|---|---|---|---|---|---|---|---|---|---|---|---|---|
| Attack↓ | TPR↑ | FPR↓ | F1↑ | TPR↑ | FPR↓ | F1↑ | TPR↑ | FPR↓ | F1↑ | TPR↑ | FPR↓ | F1↑ | TPR↑ | FPR↓ | F1↑ | TPR↑ | FPR↓ | F1↑ | TPR↑ | FPR↓ | F1↑ |
| BadNets | 90.06 | 0.03 | 94.65 | 90.06 | 0.03 | 94.65 | 90.06 | 0.03 | 94.65 | 90.06 | 0.03 | 94.65 | 90.06 | 0.03 | 94.65 | 90.06 | 0.03 | 94.65 | 90.06 | 0.03 | 94.65 |
| Blended | 94.65 | 0.175 | 96.385 | 94.65 | 0.175 | 96.385 | 94.65 | 0.175 | 96.385 | 90.74 | 0.115 | 94.38 | 90.74 | 0.115 | 94.38 | 90.74 | 0.115 | 94.38 | 90.74 | 0.115 | 94.38 |
| LF | 99.8 | 0.07 | 99.6 | 99.8 | 0.07 | 99.6 | 99.8 | 0.07 | 99.6 | 99.8 | 0.07 | 99.6 | 99.8 | 0.07 | 99.6 | 99.8 | 0.07 | 99.6 | 99.8 | 0.07 | 99.6 |
| SIG | 100 | 0.04 | 99.66 | 100 | 0.04 | 99.66 | 100 | 0.04 | 99.66 | 100 | 0.04 | 99.66 | 99.44 | 3.99 | 72.28 | 99.44 | 3.99 | 72.28 | 99.44 | 3.99 | 72.28 |
| SSBA | 99.62 | 0.04 | 99.63 | 99.62 | 0.04 | 99.63 | 99.62 | 0.04 | 99.63 | 99.62 | 0.04 | 99.63 | 99.62 | 0.04 | 99.63 | 99.62 | 0.04 | 99.63 | 99.62 | 0.04 | 99.63 |
| TACT | 100 | 0.07 | 99.68 | 100 | 0.07 | 99.68 | 100 | 0.07 | 99.68 | 100 | 0.07 | 99.68 | 100 | 0.07 | 99.68 | 100 | 0.07 | 99.68 | 100 | 0.07 | 99.68 |
| WANet | 97.8 | 0.31 | 97.4 | 97.8 | 0.31 | 97.4 | 97.8 | 0.31 | 97.4 | 97.8 | 0.31 | 97.4 | 97.8 | 0.31 | 97.4 | 97.8 | 0.31 | 97.4 | 97.8 | 0.31 | 97.4 |

### D.2 COMPARISON BETWEEN CVBT AND VARIANCE

To compare the capabilities of CVBT and variance on identifying the target class, we provide the values of CVBT and variance for different labels under various attacks and also present the Z-score estimate for the target label in Fig. 12. The disparity between CVBT and circular distribution variance

is notably pronounced within the target category while on other clean labels, CVBT is only slightly larger than variance. This also results in the Robust Z-score corresponding to CVBT, the statistical measure for detecting the target category, being greater than the value corresponding to variance, further proving that CVBT more accurately reflects the anomalies of the target category compared to variance.

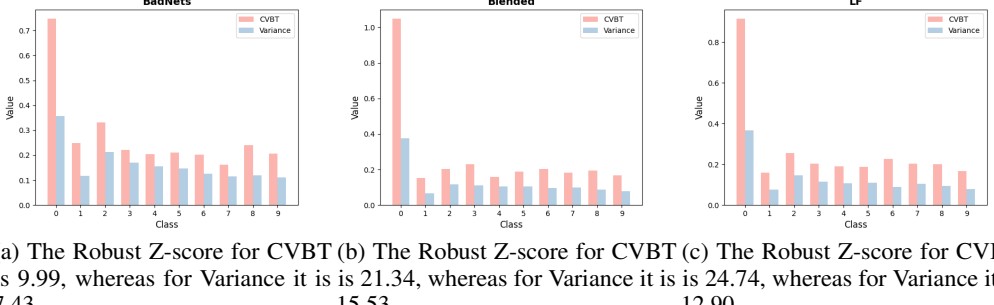

(a) The Robust Z-score for CVBT is 9.99, whereas for Variance it is 7.43. (b) The Robust Z-score for CVBT is 21.34, whereas for Variance it is 15.53. (c) The Robust Z-score for CVBT is 24.74, whereas for Variance it is 12.90.

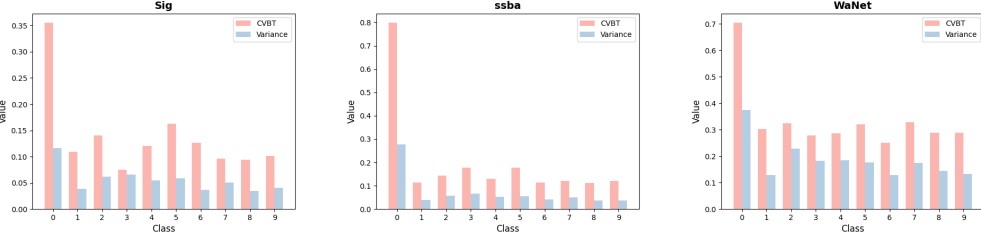

(d) The Robust Z-score for CVBT is 8.44, whereas for Variance it is 3.33. (e) The Robust Z-score for CVBT is 37.50, whereas for Variance it is 15.94. (f) The Robust Z-score for CVBT is 13.04, whereas for Variance it is 7.90.

Figure 12: CVBT and circular distribution variance on CIFAR-10 with PreAct-ResNet18 across different labels.

### D.3 COMPARISON BETWEEN COSINE DISTANCE AND RADIAN

In this section, we compare the different estimation methods of the sample closeness metric. In Tab. 15 when we replaced $1 - \cos(\theta)$ in the sample closeness metric with $\theta$, there was no significant change in the True Positive Rate (TPR) across different attacks. However, the False Positive Rate (FPR) significantly increased under certain attacks. This is because when $\theta$ approaches $\theta$ degrees, the derivative of the cosine distance is less than 1. This implies that the differences in $\theta$ are greater than those in cosine distance, resulting in the detection of some boundary points within clean samples.

Table 15: The detection performance of AGPD with radian and cosine distance. The results are evaluated on Preact-ResNet18.

| Attack | AGPD-Theta | | | AGPD | | |
|---|---|---|---|---|---|---|
| | TPR↑ | FPR↓ | F1↑ | TPR↑ | FPR↓ | F1↑ |
| BadNet | 87.14 | 0.16 | 92.40 | 90.06 | 0.03 | 94.65 |
| Blended | 99.84 | 0.00 | 99.91 | 99.98 | 0.02 | 99.88 |
| LF | 99.50 | 0.01 | 99.69 | 99.80 | 0.07 | 99.60 |
| SSBA | 99.44 | 0.02 | 99.64 | 99.62 | 0.04 | 99.63 |
| WaNet | 94.77 | 26.88 | 41.69 | 97.80 | 0.31 | 97.40 |
| Input-Aware | 97.59 | 18.08 | 52.42 | 88.25 | 1.13 | 88.61 |
| TaCT | 99.68 | 0.00 | 99.82 | 100.00 | 0.07 | 99.68 |
| Adap-Blend | 90.00 | 1.07 | 90.15 | 89.32 | 0.33 | 92.89 |

## D.4 ANALYSIS OF THE DISCRIMINATIVE DEGREE OF ACTIVATION GRADIENT.

### D.4.1 GCDS FOR VARIOUS ATTACKS ON CIFAR-10

Fig. 13 and Fig. 14 present the GCDs for all classes in CIFAR-10 under eight backdoor attacks with 10% poisoning ratio, based on the model structures of Preact-ResNet18 and VGG19-BN, respectively. It can be noticed that the target class (covering both black and blue arcs) occupies a longer arc on the circle compared with the clean classes across different model structures and backdoor attacks. Note that we moved all clean classes' arcs to different areas on the circle to avoid visual overlap.

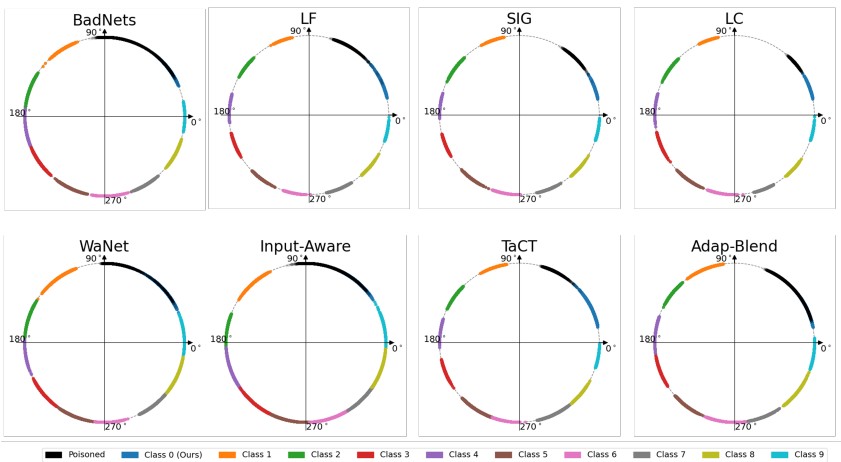

Figure 13: Gradient circular distributions (GCDs) of multiple backdoor attacks on CIFAR-10 with Preact-ResNet18.

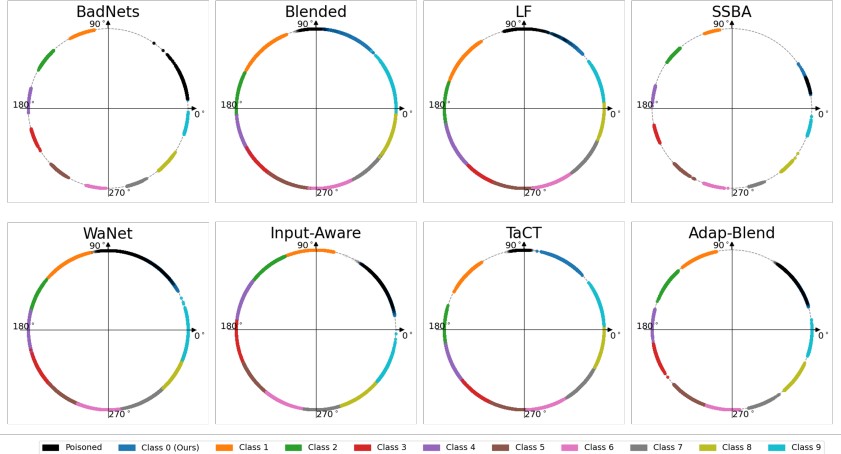

Figure 14: Gradient circular distributions (GCDs) of multiple backdoor attacks on CIFAR-10 with VGG19-BN.

### D.4.2 THE SEPARATION BETWEEN POISONED AND CLEAN SAMPLES

Fig. 15 displays the cosine similarities of samples in the target class with a clean basis, which are computed in activation gradient and activation spaces, respectively. We exhibit the distributions of cosine similarities of four convolutional layers. If a sample is clean, the cosine similarity should be large. As shown in Fig. 15, the separation of the distribution of cosine similarities between clean and poisoned samples is larger in the activation gradient space, which can be observed in many convolutional layers.

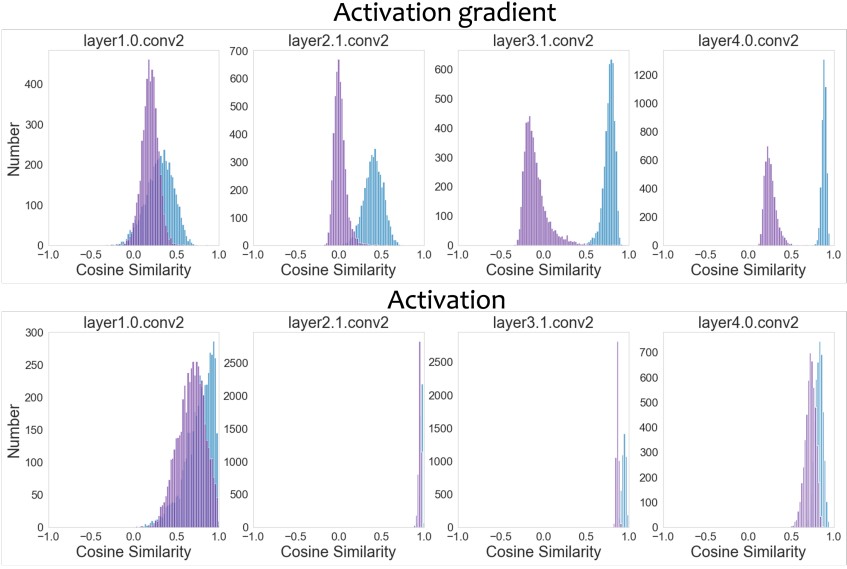

Figure 15: The distribution of the cosine similarities of samples from the target class with a clean sample in multiple convolutional layers. The model structure is Preact-ResNet18. The purple represents poisoned samples, while the blue represents clean ones.

### D.4.3 ACTIVATION VS. ACTIVATION GRADIENT.

Table 16: Silhouette scores of the target class under eight attacks, measured in activation and activation gradient spaces, using Preact-ResNet18.

|  | BadNets | Blended | LF | SSBA |
|---|---|---|---|---|
| Activation | 0.529 | 0.485 | 0.472 | 0.497 |
| Activation Gradient | 0.664 | 0.696 | 0.610 | 0.623 |
|  | WaNet | Input-Aware | TaCT | Adap-Blend |
| Activation | 0.544 | 0.457 | 0.379 | 0.403 |
| Activation Gradient | 0.605 | 0.466 | 0.515 | 0.491 |

To demonstrate that the separations of clean and poisoned samples differ in activation gradient and activation spaces, we show the distribution of cosine similarities between samples and the clean basis from both spaces. Due to the space limitation, we provide the results in Appendix D.4.2. To quantify the separation, we utilize Silhouette Score (Rousseeuw, 1987) to measure the distance between two clusters in both spaces across all convolutional layers. The range of the Silhouette Score is between -1 and 1, with higher values indicating better separability between the two clusters. Considering that different depths of convolutional layers correspond to different separations, Tab. 16 presents the maximum Silhouette Scores among all convolutional layers for eight backdoor attacks, from which we find that Silhouette Scores are larger in activation gradient space.

### D.5 COMPUTATION OVERHEAD

Tab. 17 illustrates the computation complexity and time (based on RTX A5000 GPU) of AGPD and the compared detection method under eight backdoor attacks with 10% poisoning ratio on CIFAR-10. We record the average of running time with standard deviation in bracket.

The cost of AGPD (excluding the model training cost) is $O((F + B + D)N)$, with $F, B, D$ indicating the cost of one forward pass, the cost of one backward pass along the trained model, and the feature dimension, respectively, as shown in Tab. 17. This cost is at the similar level with most compared methods. Besides, compared with model training, which cost is $O(T \times (F + B) \times N)$ with $T$ indicating the training epochs, the additional cost of AGPD is negligible.

Table 17: Computation complexity and time of AGPD and the compared methods on CIFAR-10. The value in the bracket represents the standard deviation. General setting: Epoch $E$; Samples $N$; Perturbation samples $N_p$; Feature Dimension $D$; Class $K$; Forward $F$; Backward $B$

| | AC | Beatrix | SCAn | Spectral | STRIP | ABL | CD | ASSET | AGPD |
|---|---|---|---|---|---|---|---|---|---|
| Complexity | $O((F+D)N)$ | $O((D^3+F)N)$ | $O((F+D)N)$ | $O(NF+K(N/K)^2 D)$ | $O((N_pF+K)N)$ | $O((F+B)EN)+O(FN)$ | $O((F+B)EN)$ | $O((3*F+2*B)EN)+O(FN)$ | $O((F+B+D)N)$ |
| Time (minute) | 1.02(0.01) | 8.92(0.38) | 1.27(0.02) | 1.73(0.06) | 3.29( 0.02) | 10.06(0.09) | 20.42(0.02) | 206.43 (12.63) | 5.02(0.03) |

## D.6 STABILITY OF AGPD

To ensure the stability of our method for clean samples during poisoned sample detection, we randomly selected 30 random seeds to choose samples from the test dataset. Tab. 18 presents the mean and standard deviation of the results from these 30 experiments, demonstrating that our method is robust to the clean dataset and is not affected by the samples from the clean dataset.

Table 18: The detection performance of AGPD with different sets of clean samples (mean±std) on CIFAR-10. The results are evaluated on Preact-ResNet18.

| Dataset | Attack | AGPD | | |
|---|---|---|---|---|
| | | TPR↑ | FPR↓ | F1 score↑ |
| CIFAR-10 | BadNets | 93.20±2.3 | 0.04±0.02 | 95.80±0.45 |
| | Blended | 99.98±0.01 | 0.02±0.01 | 99.83±0.06 |
| | LF | 99.8±0.04 | 0.07±0.01 | 99.61±0.05 |
| | SSBA | 99.68±0.14 | 0.05±0.01 | 99.61±0.05 |
| | WaNet | 97.77±0.36 | 0.26±0.11 | 99.63±0.06 |

## E THE RESULTS OF THE COMPARED ASSET ON RESNET18

In our above experiment, we found that detection performance of ASSET is deviated from the results reported in the paper (Pan et al., 2023), such as BadNets and Blended attacks. Therefore, we decided to replicate these detection results using the recommended model structure, ResNet18. We tested six different attacks, including BadNets, Blended, WaNet, Input-Aware, TaCT, and Adap-Blend. The experiments were conducted on the CIFAR-10 dataset with 10% poisoning ratio. The corresponding results are shown in Tab. 19. When changing the model structure from Preact-ResNet18 to ResNet18, we discovered that the detection performance of ASSET becomes better, such as the TPRs of BadNets, Blended, and TaCT. However, when the form of the trigger is too complex and dynamic, which requires the model to spend more epochs learning it, this poses a greater challenge for the loss-based method, like ABL and ASSET. The work (Wu et al., 2022) provides more analysis of quick learning of backdoors.

Table 19: The results of ASSET on CIFAR-10 with ResNet18.

| | BadNets | Blended | WaNet | Input-Aware | TaCT | Adap-Blend |
|---|---|---|---|---|---|---|
| TPR | 90.70 | 99.90 | 1.88 | 0.25 | 100.00 | 54.58 |
| FPR | 0.18 | 9.43 | 1.69 | 0.22 | 0.00 | 38.90 |
| F1 | 94.32 | 69.97 | 3.55 | 0.48 | 100.00 | 23.43 |

## F T-SNE RESULTS

In this section, we provide the t-SNE visualizations of ten backdoor attacks conducted on CIFAR-10 with Preact-ResNet18 in Fig. 16. The activations are extracted from the last convolutional layer (*layer4.1.conv2*). From Fig 16, it is evident that the dispersion of activations of poisoned and clean samples is significant in some attacks, such as BadNets and Blended. Consequently, most activation-based methods perform well against these attacks. Additionally, the dispersion of activations of the target class is relatively small in CTRL attacks, where these methods fail.

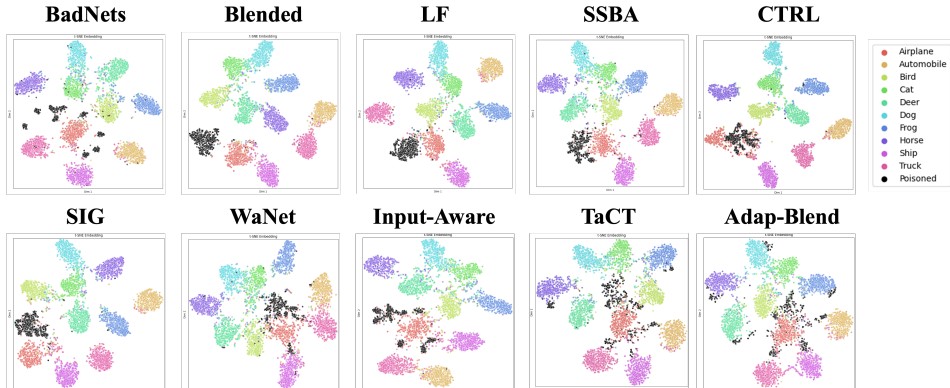

Figure 16: t-SNE visualization of ten backdoor models trained on poisoned CIFAR-10 under 10% poisoning ratio for non-clean label attacks and 5% poisoning ratio for clean label attacks. The model architecture is Preact-ResNet18.

# G   RESULTS FOR ADAPTIVE ATTACKS

In this section, we present the results and analysis for adaptive attacks in which adversaries also added perturbations to clean samples used by defenders. The performance of AGPD under these conditions is summarized in Tab.20. Additionally, we visualize the distribution of cosine similarities among poisoned, noisy, and clean samples in the activation gradient space under two modes, as shown in Fig.17 and Fig.18, respectively, to further demonstrate the effectiveness of our method.

**Experimental setup:**   The poisoned dataset consists of 10% poisoned samples and 10% clean samples with added noise. In these clean samples, noise is inserted into the images of the blending ratio 0.2 without altering the labels. There are two modes of adding noise: in the **Random mode**, all noise is generated randomly, while in the **Fixed mode**, half of the noise is fixed, and the other half is generated randomly. We evaluated the effectiveness of AGPD in detecting various attacks on the CIFAR-10 dataset using the PreactResNet-18 architecture.

**Analysis:**   we observed that in both two noise modes, the distribution of noise samples closely resembles that of clean samples. Despite these samples with noise, they can retain their original features, resulting in similar gradients to those of clean samples. This observation underscores the robustness of AGPD in distinguishing between clean and poisoned samples, even when clean samples are added with noise. The preservation of original features in noisy samples ensures that their gradients remain unaffected, allowing AGPD to effectively detect poisoned samples without interference from noise. Additionally, we offered the performance of AGPD against adaptive attacks in Tab. 20. It can be seen that the proposed method maintained high TPR and relatively low FPR across different backdoor attacks and both noise-adding modes.

Table 20: Performance of AGPD against adaptive attacks with different attacks under poisoning ratio=10% on CIFAR-10 and ResNet-18.

| Attack | Mode | TPR | FPR | F1 |
|---|---|---|---|---|
| BadNets | Random | 86.68 | 0.01 | 92.83 |
| | Fixed | 87.24 | 0.18 | 85.42 |
| Blended | Random | 99.88 | 0.00 | 99.93 |
| | Fixed | 99.56 | 0.00 | 99.76 |
| LF | Random | 99.24 | 0.01 | 99.59 |
| | Fixed | 99.36 | 0.01 | 99.62 |
| SSBA | Random | 99.42 | 0.03 | 99.56 |
| | Fixed | 99.52 | 0.03 | 99.63 |

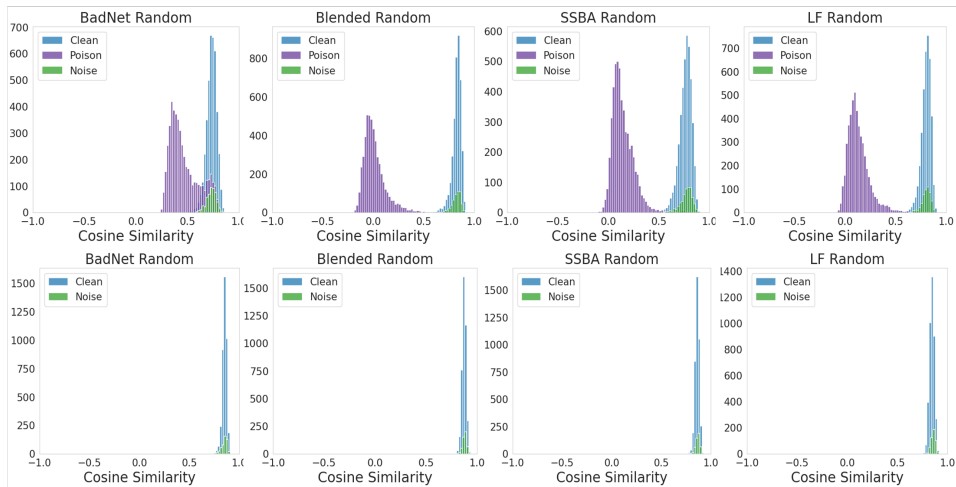

Figure 17: The distribution of the cosine similarities of samples from the target class (**above**) and samples from one non-target class (**below**) in activation space for Random mode across multiple convolutional layers. The purple represents poisoned samples, the blue represents clean ones, and the green represents noise ones.

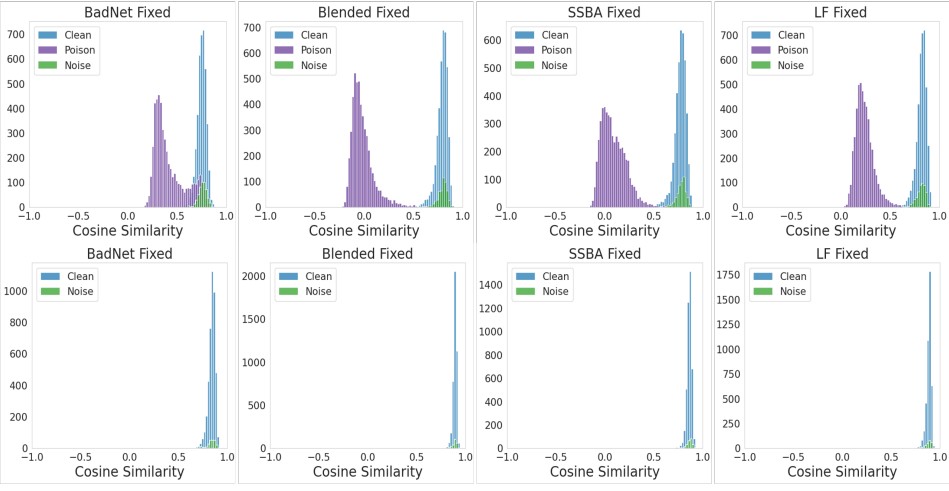

Figure 18: The distribution of the cosine similarities of samples from the target class (**above**) and samples from one non-target class (**below**) in activation space for Fixed mode across multiple convolutional layers. The purple represents poisoned samples, the blue represents clean ones, and the green represents noise ones.

# H   RESULTS FOR CLEAN-LABEL ATTACKS

In this section, we provide a visualization of the cosine similarity distribution for samples in the target class to illustrate AGPD's detection capability against clean-label attacks, as shown in Fig.19. It is evident that there is a clear separation between the two types of samples in the activation gradient space, even though the poisoned and clean samples belong to the same class. Clean-label attacks, despite exhibiting similar visual features in the input space, inherently disrupt the internal characteristics of the original samples within the model. This disruption impacts the learning of clean features and results in the model memorizing the adversarial triggers. This memorization causes a divergence in the gradient space between clean and poisoned samples, underscoring the effectiveness of our AGPD method in detecting clean-label attacks.

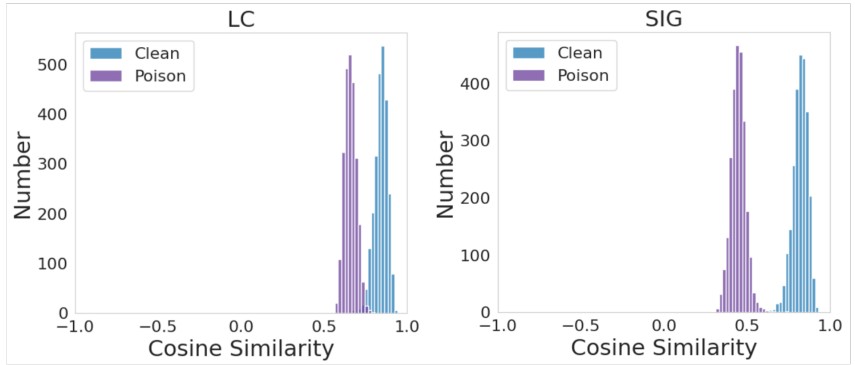

Figure 19: The distribution of the cosine similarities of samples from the target class.

# I    RESULTS FOR NOISY AND POISONED SAMPLES

Considering adversaries are able to inject noisy samples to further improve the attack stealthiness, we provide the visualization of the distribution of cosine similarities for noisy and clean samples sharing the same label under three backdoor attacks in Fig.20.

**Experimental setup:**   We create a noisy sample by adding noise to a poisoned sample and reverting it to its original label.

**Analysis:**   We found that the gradient distributions of noisy samples and clean samples are almost identical. This is because the noisy sample retains the class information and has not been altered to the target label, so the inherent clean information of the model predominantly influences the gradient.

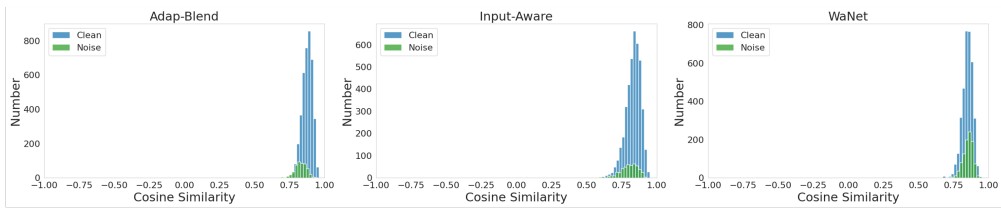

Figure 20: The visualization of cosine similarities for noisy and clean samples. The blue represents the clean ones and the green represents noisy ones.

