# OpenReview forum: "Activation Gradient based Poisoned Sample Detection Against Backdoor Attacks"
_ICLR.cc/2025/Conference — ICLR 2025 Poster_

### Official Review · Reviewer_AHKK · 2024-10-28

**Soundness:** 3
**Presentation:** 3
**Contribution:** 3
**Rating:** 6
**Confidence:** 4

**Summary:**

With the inspiration that a backdoored model is apt to map significantly different poisoned samples and clean samples of the backdoor target to similar activation areas, this paper introduces a novel measurement, i.e., the circular distribution of the gradients w.r.t sample activation, namely GCD, which works to identify the target class of the backdoor and consequently separate poisoned and clean samples within the target class. Accordingly, this paper proposes a sample detection approach called AGPD to achieve dataset purification. Extensive experiments show AGPD's advanced performance in detecting and isolating poisoned samples.

**Strengths:**

- This paper proposes a novel measurement to show the dispersion of poisoned and clean samples in the model's activations

- This paper is well-written and enforces a high readability with variant and helpful illustrations

- This paper comprehensively evaluates AGPD's performance by considering diverse poisoning attacks, comparing with different related defenses, and conducting experiments on variant datasets.

- This paper considers the ablation study on different hyper-settings of APGD and involves two reasonable adaptive attacks that have considered APGD's defense w.r.t the GCD distribution.

**Weaknesses:**

(1) The use of $arccos(\cdot)$ in Eq.(2) is ambigious as $cos\(\cdot\)$ is used in Eq.(3) and Eq.(4) as well.

(2) The calculation of GCD across all model layers is not clearly formulated for each sample.

(3) The influence of the clean sample by random sampling for the basic sample pair is not studied.

(4) There is lacking the rationale for setting the hyper-parameter $\tau_z = e^2$.

(5) For all-to-all attacks, a better explanation of how the dataset separation by APGD is achieved is required.

(6) The description of Adpative-Blend in Appendix B is incomplete and different from the default attack setting.

(7) It is unclear how the noisy and poisoned samples are separated against WaNet and Adaptive-Blend attacks.

---
In the following, further questions related to the above points are detailed with the notion of "Q-#".

**Questions:**

Q-(1): The calculation of $\rho_{x_0}\(\mathcal{D}\)$ in Eq.(3) and $s_{x_0}\(x_i\)$ in Eq.(4) basically transforms e.g. $\theta_{x_0}\(x_i\)$ back to $\frac{g\(x_i\) \cdot g\(x_0\)}{||g\(x_i\)|| ||g\(x_0\)||}$. By simplifying the formulation in Eq.(3) and Eq.(4), does this mean it is essentially calculating the cosine similarity of the gradients between the basis pair and each input sample? If this is indeed the case, is the calculation of radian in Eq.(2) useful merely for the visualization in figures like Figure 1.?

Q-(2): In section 3, the channel-wise activation gradient $g^{\(l\)}$ is first introduced in Eq.(1). However, the formulating w.r.t the index of the model layer is missing in the consequent methodology description. Can the authors elaborate on how the channel-wise gradients in each layer are merged into the final GCD of an input sample $x_i$?

Q-(3): The basis pair is crucial to disperse samples; thus, a small clean set is necessary for APGD detection. As even a single clean sample per class has been proven effective, how does the selection of clean samples influence APGD's performance? Or is APGD resistant to any random sampling of clean samples?

Q-(4): Different from the study on the threshold $\tau_s$ in section 5.4, $\tau_z$ remains equal $e^2$. What is the rationale of setting $\tau_z = e^2$?

Q-(5): In Figure 4, APGD detection on the target class works better against all-to-one attacks than all-to-all attacks. While detecting poisoned samples of all-to-all attacks, does APGD handle the separation of all-to-all attacks class by class via the implementation of Stage 3?

Q-(6): In the default setting of Adaptive-Blend, the poisoning ratio is relatively low (poisoned/total = 150/50000 = 0.3% for CIFAR10), where the same amount of noisy samples as poisoned samples are added by setting the conservatism ratio = 1.0, which is unfortunately missing in Table 7. Moreover, considering the default poisoning ratio = 5%, the ASR of 66.17% for Apat-Blend is much lower than that of the original paper. Does this imply that the Adptive-Blend in this paper has some biases from the default attack configuration?

Q-(7): Considering that both WaNet and Adaptive-Blend attacks involve noisy samples to improve the attack stealthiness, can the authors provide some visualization on the separation of APGD between the poisoned samples and noisy samples in both attacks?

---

> ### Author Response · Authors · 2024-11-22
> **Response to Reviewer AHKK (Part 1/2)**
>
> Dear Reviewer AHKK,
>
> We are truly thankful for the reviewer’s positive feedback.
>
> **Q1. The reason for using $\arccos$ in Eq.(2), while employing $\cos$ in Eq.(3) and Eq.(4).**
>
> **R1:** Thank you! We explain the reason for the definition of Gradient Circular Distribution (GCD) and various metrics.
>
> * **Equivalence from a computational perspective:** From a mathematical standpoint, Eq.(3) and (4) indeed directly compute the cosine similarity of the gradients between the basis pair and each input sample.
> * **Usefulness of the definition of Gradient Circular Distribution:** The definition of Gradient Circular Distribution (GCD) provides a new perspective based on circular distribution to analyze poisoned and clean samples. The CVBT and Sample-level Closeness Metric are one of effective quantitative indicators we propose based on this distribution. Additional metrics can be developed based on circular distribution to analyze backdoor attacks.
>
> **Q2. Can the authors elaborate on how the channel-wise gradients in each layer are merged into the final GCD of an input sample？**
>
> **R2:** Thank you for your suggestions.
>
> * **The relationship between GCD and layer:** In Section 3.1 and 3.2 of our revised version, we provide a more clear definition of the Gradient Circular Distribution (GCD) and explain the relationships between different layers. Specifically, for each layer, we calculate the corresponding GCD and metrics.
> * **How our method (AGPD) utilizes layers:** In Section 4.2 of our main manuscript, we first utilize the CVBT and the corresponding Z-score obtained for each layer to select the layer with the most significant differences. We then use the GCD of this specific layer for detecting poisoned samples.
>
> **Q3. Is APGD resistant to any random sampling of clean samples?**
>
> **R3:** Thank you for your question! AGPD is less affected by the randomness of clean samples, as can be seen from the following points:
>
> * **Target class detection is not affected**: In Figure 4 of our main manuscript (see Section 5.3), we analyze the CVBT and Z-score of the target class under different attacks, where clean samples are randomly selected. Although there is some randomness in the CVBT of different classes, the CVBT of the target class significantly exceeds that of other classes during detection. This results in the Z-score for the target class far exceeding the threshold corresponding to the confidence level of $\text{1e-13}$ in the hypothesis test.
> * **Poisoned sample detection is not affected**: To ensure the stability of our method for clean samples during poisoned sample detection, we randomly selected 30 random seeds to choose samples from the test dataset. **Table 1** presents the mean and standard deviation of the results from these 30 experiments, demonstrating that our method is robust to the clean dataset.
>
> **Table 1: Performance of AGPD with different sets of clean samples (mean±std).**
>
> | Attack  | TPR          | FPR         | F1 score     |
> | ------- | ------------ | ----------- | ------------ |
> | BadNets | 93.20%±2.3%  | 0.04%±0.02% | 95.80%±0.45% |
> | Blended | 99.98%±0.01% | 0.02%±0.01% | 99.83%±0.06% |
> | LF      | 99.8%±0.04%  | 0.07%±0.01% | 99.61%±0.05% |
> | SSBA    | 99.68%±0.14% | 0.05%±0.01% | 99.61%±0.05% |
> | WaNet   | 97.77%±0.36% | 0.26%±0.11% | 99.63%±0.06% |
>
> **Q4. What is the rationale for setting the hyper-parameter $\tau_z$?**
>
> **R4:** Thanks! We would like to address the above concern as follows:
>
> * **Guideline for setting $\tau_z$:** We recommend referring to the work [1], which provides guidelines on determining its value which is used to detect outliers from the perspective of hypothesis testing.
>   * In statistical hypothesis testing, **the significance level**, often denoted by $\alpha$, is a critical concept that represents the probability of committing a Type I error—rejecting a true null hypothesis. When using a Z-score to test hypotheses, this threshold corresponds to a specific value on the standard normal distribution.
>   * **The Z-score threshold** is determined such that the probability of observing a value as extreme, or more extreme, is equal to the significance level $\alpha$.
>   * **In AGPD**, $\tau_z$ is set to $e^2$, aligning with a very stringent confidence level of approximately $\text{1e-13}$. This setup ensures the target class identified are exceptionally rare under the null hypothesis. This threshold is much stricter than typical significance levels, reducing the probability of false positives to an extreme degree.
> * **Robustness to $\tau_z$:** As shown in Fig. 4 in the manuscript, we record the Z-score values for all classes in 160 all-to-one attacks. It is observed that the anomaly value $z$ of the target class is much higher than those of all clean classes, implying that there is a wide range for setting a suitable value of $\tau_z$.
>
> References:
>
> [1] Boris Iglewicz and David C Hoaglin. Volume 16: how to detect and handle outliers. Quality Press, 1993.

---

> ### Author Response · Authors · 2024-11-22
> **Response to Reviewer AHKK (Part 2/2)**
>
> **Q5. The requirement of more explanation of the mechanism of Stage 3 in all-to-all attacks.**
>
> **R5:** Thanks for your suggestion. We would like to take this chance to further clarify the mechanism of AGPD on handling all-to-all attacks.
>
> * **AGPD can handle both all-to-one and all-to-all attacks, and even multi-target attacks simultaneously.** As illustrated in Line 245-246, "*if $z_k^{(l^\*)}$ exceeds a threshold $\tau_z$ , i.e., $z_k^{(l^\*)} \geq \tau_z$, $k$ is identified as a target class, otherwise clean class.*" **The reationale of this mechanism is that as long as there are poisoned samples in one class, then its dispersion score $z_k^{(l^\*)}$ will be large**, and be much larger than the dispersion score of clean class. It is independent of the number of potential target classes. Thus, we claim that AGPD can handle any kind of attacks simultaneously. According to the reported results in Tables 1 and 2, there are no significant performance difference among all-to-one, all-to-all, and multi-target attack scenarios.
> * **Yes, AGPD handles the separation of all-to-all attacks class by class via the implementation of Stage 3.** Given any identified target class $k^*$, then we conduct Stage 3 to filtering poisoned samples from that class. The filtering operations on different classes are independent.
>
> **Q6. Does the results of Adaptive-Blended in Appendix B imply that the Adptive-Blend in this paper has some biases from the default attack configuration?**
>
> **R6:** Thank you for your valuable suggestions. We would like to address your concerns with the following points:
>
> * **At low ratios, our AGPD method effectively detects Adaptive-Blend attacks.** As shown in **Table 2**, our approach outperforms other feature-based detection methods. Specifically, the silhouette value for our gradient-based detection is 0.3258, compared to 0.2271 for other feature-based methods. The silhouette value indicates the separation quality within classes, with higher values signifying better separation. Since the poisoned and clean samples belong to different classes but are assigned the target label, the gradient remains less affected by noisy samples. In contrast, noisy samples weaken the backdoor signal, reducing the difference between the feature of poisoned and clean samples, as noted in [1].
> * **Impact of high ratios on ASR:** At higher ratios, the issue of noisy samples significantly impacts the Attack Success Rate (ASR). Our experiments followed the original paper's 1:1 ratio of poisoned to noisy samples. An excessive number of noisy samples negatively affects the ASR, as discussed in Table 2 of [1], where an increase in noisy samples correlates with a decrease in ASR.
>
> **Table 2: Performance of AGPD and other detection methods with Adapt-Blend under poisoning ratio=0.3\%.**
>
> |          | TPR    | FPR    | F1     |
> | -------- | ------ | ------ | ------ |
> | AC       | 0.00%  | 0.00%  | 0.00%  |
> | Beatrix  | 0.67%  | 5.95%  | 0.05%  |
> | SCAn     | 0.00%  | 0.00%  | 0.00%  |
> | Spectral | 0.67%  | 15.05% | 0.02%  |
> | STRIP    | 0.67%  | 9.19%  | 0.03%  |
> | AGPD     | 92.00% | 0.69%  | 43.81% |
>
> References:
>
> [1] Qi X, Xie T, Li Y, et al. Revisiting the assumption of latent separability for backdoor defenses[C]//The eleventh international conference on learning representations. 2023.
>
> **Q7. An explanation of how the noisy and poisoned samples are separated against WaNet and Adaptive-Blend attacks.**
>
> **R7:** Thank you very much for your suggestions.
>
> * **A noisy sample is created by adding noise to a poisoned sample and reverting it to its original label.** Since noisy samples and poisoned samples have different labels, AGPD does not consider the relationship between these samples during detection. In our revised version, we included gradient distributions of noisy samples and clean samples with other label. We observed that the gradient distributions of noisy samples and clean samples are almost identical. This is because the noisy sample retains the class information and has not been altered to the target label, so the inherent clean information of the sample influences the gradient.
>
> * **Noisy samples have limited effectiveness in enhancing attack stealthiness at high poisoning ratios.** In Table 1 of our main manuscript, both gradient-based AGPD and feature-based SCAn successfully detect WaNet, Adaptive-Blend, and Input-Aware attacks. As mentioned in [1], the stealthiness of poisoned samples decreases as the poisoning ratio increases.
>
> References:
>
> [1] Qi X, Xie T, Li Y, et al. Revisiting the assumption of latent separability for backdoor defenses[C]//The eleventh international conference on learning representations. 2023.

---

> > ### Comment · Reviewer_AHKK · 2024-11-25
> > **Thank you for the response**
> >
> > Dear authors,
> >
> > Thank you for your effort in the response.
> >
> > After carefully reading the rebuttal, many of my concerns (i.e., Q1-Q5) are addressed, which further improve the clarity of the method AGPD and demonstrate its effectiveness against variant attacks. Nevertheless, one concern remains regarding attacks using noisy samples.
> >
> > In Table 2 of reply 6, AGPD achieves high TPR and low FPR against Adapt-blend attack with poisoning ratio $0.3\%$, but the F1 score is $43.81\%$, with which we can derive the precision equal $28.75\%$. As the authors mention in Reply 7 that noise samples and clean samples are clustered tightly in the view of cosine similarity, does the low precision imply many False Positives in the detection, i.e., many clean and(or) noisy samples are detected as poisoned samples? How does such a detection impact the final model performance compared with the baselines in the paper's Table 1?
> >
> > I would appreciate further explanation of my remaining concern above, but I acknowledge the quality of the paper at present. Therefore, I would keep my score and see the discussion with other reviewers.

---

> ### Author Response · Authors · 2024-11-25
> **Appreciation for Your Feedback**
>
> Dear Reviewer AHKK,
>
> Thanks for your positive feedback and valuable suggestions. We explain the current phenomenon.
>
> - The number of false positives among clean samples is 342, while the detected poisoned samples amount to 138. **Although the precision is relatively low, the proportion of misclassified clean samples is only 0.6% of the total.**
>
> - **The occurrence of false positives can be attributed to the extremely low poisoning rate of AGPD-Blend and the randomness of the trigger in the poisoned samples.** This results in low similarity of sample-level closeness among poisoned samples. The silhouette value of the gradient circular distribution for the target class is 0.3258, indicating the presence of some poisoned samples with weaker backdoor effects. Consequently, during Stage 3, a small number of target clean samples were incorrectly identified as poisoned samples. However, the Z-score corresponding to the target class is 9.123, suggesting that these difficult-to-identify poisoned samples constitute a minor proportion, and the rate of misclassified clean samples remains very low overall.
>
> - We evaluated the accuracy and attack success rate (ASR) after training with the filtered dataset, yielding results of **91.40%** and **1.40%**, respectively, compared to the original accuracy and ASR of **92.6%** and **71.5%**. This indicates that the final model's performance is minimally affected. In comparison to other methods, our approach not only mitigates the risks associated with retraining the model—resulting in a low ASR—but also maintains the model's original performance, as evidenced by the high accuracy.
>
> We hope this addresses your concerns, and we also welcome any further constructive feedback you might provide to enhance our submission.
>
> Best regards,
>
> Authors

---

> ### Comment · Reviewer_AHKK · 2024-11-25
> **Thank you for the timely response**
>
> Thank you for your timely response. With the updated experiment against the Adaptive-Blend attack, I can confirm the robustness and effectiveness of AGPD detection. I hold a positive view of this work and look forward to discussing the concerns raised by other reviewers.

---

> ### Author Response · Authors · 2024-11-26
> **Appreciation for Your Feedback**
>
> Dear Reviewer AHKK,
>
> Thank you for your positive feedback and valuable suggestions. We appreciate the opportunity to further discuss the effectiveness and robustness of our method, and we will include the experiments and details we discussed in the final version of the manuscript.
>
> Once again, thank you for your thorough review and positive evaluation.
>
> Best regards,
>
> The Authors

---

### Official Review · Reviewer_pwS4 · 2024-10-29

**Soundness:** 2
**Presentation:** 3
**Contribution:** 2
**Rating:** 6
**Confidence:** 4

**Summary:**

The paper presents a new approach to identifying poisoned samples in datasets used for deep neural network training. The core proposal, termed Activation Gradient-Based Poisoned Sample Detection (AGPD), leverages a novel metric called Gradient Circular Distribution (GCD) to detect discrepancies between clean and poisoned samples. Key insights include observing that in backdoor attacks, models often map poisoned samples and clean samples to similar activation regions, which AGPD detects by analyzing gradient direction distributions. This method involves three main steps: calculating GCD for each class, identifying target classes based on dispersion, and filtering out poisoned samples.

**Strengths:**

(1) Introducing GCD as a novel measure adds a unique, technical depth to the approach, potentially applicable to various attack scenarios.

(2)  AGPD shows high detection performance with minimal additional clean data, a notable advantage for real-world applicability, where such data may be limited.

**Weaknesses:**

(1) The proposed method is founded on two core observations: (i) the Gradient Circular Distribution (GCD) of the target class exhibits greater dispersion than those of clean classes, and (ii) poisoned and clean samples are distinguishable by their clear clustering in separate regions. However, it remains unclear whether these observations hold consistently for clean-label attacks, where the mapping directions of poisoned and benign samples could exhibit higher similarity given their similar input-space characteristics. Further discussion on this aspect would strengthen the paper’s analysis.

(2) The choice of baseline methods appears limited to older techniques. A broader comparison with more recent, state-of-the-art defense methods would provide a more thorough assessment of the proposed method’s performance relative to the latest advances in poisoned sample detection.

(3) Attack Success Rate (ASR) is a key metric commonly used in backdoor detection evaluations. Its omission from Table 1 raises questions about the comprehensiveness of the evaluation metrics chosen. Including ASR would provide a clearer view of the method's efficacy against backdoor attacks.

(4)   In the main evaluation, a poisoning ratio of 10% is used for non-clean label attacks and 5% for clean-label attacks. Since clean-label attacks are generally more challenging, lower poisoning ratios are typically applied to non-clean label attacks in real-world scenarios. This discrepancy between the paper’s experimental setup and actual attack contexts calls for further clarification. Additionally, practical attack scenarios often employ very low poisoning ratios (1% or 0.5%), feature separability-based defenses tend to be less effective in this setting. More discussion should be included.

**Questions:**

(1) The proposed method is founded on two core observations: (i) the Gradient Circular Distribution (GCD) of the target class exhibits greater dispersion than those of clean classes, and (ii) poisoned and clean samples are distinguishable by their clear clustering in separate regions. However, it remains unclear whether these observations hold consistently for clean-label attacks, where the mapping directions of poisoned and benign samples could exhibit higher similarity given their similar input-space characteristics. Further discussion on this aspect would strengthen the paper’s analysis.

(2) The choice of baseline methods appears limited to older techniques. A broader comparison with more recent, state-of-the-art defense methods would provide a more thorough assessment of the proposed method’s performance relative to the latest advances in poisoned sample detection.

(3) Attack Success Rate (ASR) is a key metric commonly used in backdoor detection evaluations. Its omission from Table 1 raises questions about the comprehensiveness of the evaluation metrics chosen. Including ASR would provide a clearer view of the method's efficacy against backdoor attacks.

(4)   In the main evaluation, a poisoning ratio of 10% is used for non-clean label attacks and 5% for clean-label attacks. Since clean-label attacks are generally more challenging, lower poisoning ratios are typically applied to non-clean label attacks in real-world scenarios. This discrepancy between the paper’s experimental setup and actual attack contexts calls for further clarification. Additionally, practical attack scenarios often employ very low poisoning ratios (1% or 0.5%), feature separability-based defenses tend to be less effective in this setting. More discussion should be included.

---

> ### Author Response · Authors · 2024-11-22
> **Response to Reviewer pwS4 (Part 1/2)**
>
> Dear Reviewer pwS4,
>
> At the beginning, we want to express our sincere gratitude to the reviewer for the positive feedbacks. We appreciate your recognition of the novelty and effectiveness of our method.
>
> **Q1. Whether the two observations hold consistently for clean-label attacks?**
>
> **R1:** Thank you for your insightful suggestions. We would like to address your points as follows:
>
> * **Characteristics of clean-label attacks:** Clean-label attacks, despite exhibiting similar visual features in the input space, inherently disrupt the internal characteristics of the original samples within the model. As noted in [1], this disruption impacts the learning of clean features and results in the model memorizing the triggers. This disruption causes a divergence in the gradient space between clean and poisoned samples.
> * **Gradient space analysis and detection capability of AGPD:** We have included detailed visualizations of the gradient space distributions for both poisoned and clean samples under two different clean-label attack scenarios, as shown in Figure 19 in the revised version (see Appendix H). These visualizations clearly demonstrate a distinct separation between the two types of samples in the gradient space, underscoring the effectiveness of our AGPD method in detecting clean-label attacks.
>
> References:
> [1]Turner A, Tsipras D, Madry A. Label-consistent backdoor attacks[J]. arXiv preprint arXiv:1912.02771, 2019.
>
> **Q2.The results of the comparison with recent state-of-the-art defense methods.**
>
> **R2:** Thanks! We are delighted to include the assessment of our method alongside recent and state-of-the-art defense methods, such as CT [1] and IBD-PSC [2], to further enhance our submission. The key insight of CT is that a post-attack model trained on a confusion dataset fails to recognize most clean samples but still correctly predicts most poisoned samples. The principle behind IBD-PSC is that the prediction confidences of poisoned samples have parameter-oriented scaling consistency.
>
> We reproduced these works using the official code provided for CT and IBD-PSC. To ensure a fair comparison, we maintained experimental settings of attacks (*i.e.*, learning rate and training epoch $E$) consistent with our main experiment, as introduced in Sec 5.1. The details are shown as follows:
>
> * **Dataset and model architecture:** We compare AGPD with CT and IBD-PSC on CIFAR-10 dataset and PreAct-ResNet18.
> * **Attack method and poisoning ratio:** We choose four classical attack methods which are BadNets, Blended, SSBA, and WaNet. The poisoning ratio is 10%.
> * **Hyper-parameters for CT:** We set the number of clean samples is 2,000 and confusion factor $\lambda$ is 20. Besides, we set confusion iteration $K=6$, confusion iter $m=6000$, a learning rate of 0.001, and the distillation ratios $\\{r_1=\frac{1}{2}, r_2=\frac{1}{5}, r_3=\frac{1}{25}, r_4=\frac{1}{50}, r_5=\frac{1}{100}\\}$.
> * **Hyper-parameters of IBD-PSC:** The number of clean samples is 2,000. The hyper-parameters for error rate, threshold $T$, and amplifying coefficient for the selected BN layer set to 0.6, 1.5, and 0.9, respectively.
>
> **Experimental results and analysis**
> The experimental results are shown in **Table 1**. It can be seen that both our method and the compared methods achieve high performance in detecting poisoned samples. Notably, our method performs well in minimizing misclassification of clean samples during the detection task.
>
> **Table 1: Performance (%) of AGPD with two compared defense methods on CIFAR-10 and PreactResNet-18.**
>
> |         |   CT    |       |        | IBD-PSC |        |        |  AGPD  |       |        |
> | ------- | :-----: | :---: | :----: | :-----: | :----: | :----: | :----: | :---: | :----: |
> |         |   TPR   |  FPR  |   F1   |   TPR   |  FPR   |   F1   |  TPR   |  FPR  |   F1   |
> | Badnet  | 95.94%  | 0.01% | 97.90% | 99.82%  | 11.02% | 64.14% | 90.06% | 0.03% | 94.65% |
> | Blended | 99.96%  | 1.36% | 94.21% | 99.47%  | 11.42% | 63.17% | 99.98% | 0.02% | 99.88% |
> | SSBA    | 100.00% | 0.08% | 99.63% | 85.64%  | 10.86% | 60.07% | 99.62% | 0.04% | 99.63% |
> | WaNet   | 91.64%  | 0.17% | 94.89% | 96.24%  | 11.41% | 61.78% | 97.80% | 0.31% | 97.40% |
>
> [1]Qi X, Xie T, Wang J T, et al. Towards a proactive {ML} approach for detecting backdoor poison samples[C] USENIX Security Symposium.
>
> [2]Hou L, Feng R, Hua Z, et al. IBD-PSC: Input-level Backdoor Detection via Parameter-oriented Scaling Consistency[J]. arXiv preprint arXiv:2405.09786, 2024.

---

> ### Author Response · Authors · 2024-11-22
> **Response to Reviewer pwS4 (Part 2/2)**
>
> **Q3. The performance evalution metric should contain the attack success rate (ASR).**
>
> **R3:** Thanks for this construtive suggestion. We agree that ASR is a key metric for evaluating detection methods against backdoor attacks. In Appendix Table 8 (see Appendix C.1), we show the ACC and ASR for model trained on datasets filtered by AGPD. To further illustrate AGPD's effectiveness, **Table 2** compares the ASR of AGPD with other methods like SCAn, CD, and STRIP. While CD and STRIP have high true positive rates, their ASR reduction is less effective due to the easy learning of backdoor attacks. In contrast, AGPD consistently maintains a low ASR, effectively removing poisoned samples and neutralizing backdoor threats. SCAn, however, shows higher ASR in some scenarios.
>
> **Table 2: Performance of different detection methods based on the model, which is trained on the filtered dataset, on CIFAR-10 with PreactResNet-18.**
>
> |             | Origin            |                   | AGPD              |                   | SCAn              |                   | CD                |                   | STRIP             |                   |
> | ----------- | ----------------- | ----------------- | ----------------- | ----------------- | ----------------- | ----------------- | ----------------- | ----------------- | ----------------- | ----------------- |
> | Attack      | Poisoned data ACC | Poisoned data ASR | Filtered data ACC | Filtered data ASR | Filtered data ACC | Filtered data ASR | Filtered data ACC | Filtered data ASR | Filtered data ACC | Filtered data ASR |
> | BadNets     | 91.82%            | 93.79%            | 91.90%            | 1.41%             | 92.98%            | 1.11%             | 86.00%            | 92.61%            | 91.52%            | 2.01%             |
> | Blended     | 93.69%            | 99.75%            | 91.52%            | 2.11%             | 91.74%            | 30.80%            | 84.65%            | 95.74%            | 92.37%            | 94.74%            |
> | LF          | 93.01%            | 99.05%            | 92.02%            | 1.60%             | 91.13%            | 44.42%            | 91.31%            | 84.10%            | 93.46%            | 92.73%            |
> | SSBA        | 92.88%            | 97.06%            | 91.70%            | 0.90%             | 91.20%            | 9.10%             | 93.48%            | 40.98%            | 90.56%            | 87.78%            |
> | SIG(5%)     | 93.40%            | 95.43%            | 91.91%            | 0.01%             | 93.09%            | 1.18%             | 92.59%            | 88.78%            | 91.83%            | 1.51%             |
> | WaNet       | 89.68%            | 96.94%            | 89.98%            | 0.86%             | 90.80%            | 4.43%             | 10.00%            | 100.00%           | 90.55%            | 94.06%            |
> | Input-Aware | 91.35%            | 98.17%            | 91.14%            | 9.67%             | 94.08%            | 0.81%             | 92.63%            | 32.68%            | 91.96%            | 90.32%            |
> | TaCT        | 93.21%            | 95.95%            | 90.71%            | 0.55%             | 92.72%            | 0.05%             | 82.95%            | 82.50%            | 92.71%            | 90.50%            |
> | Adap-Blend  | 92.87%            | 66.17%            | 91.25%            | 4.40%             | 92.78%            | 10.06%            | 92.87%            | 66.17%            | 92.65%            | 56.90%            |
>
> **Q4. The reason about different poisoning ratio setting for non-clean label attacks and clean-label attacks, and the performance of AGPD under low poisoning ratios.**
>
> **R4:** Thank you for your insightful feedback. We would like to address your concerns from the following perspectives:
>
> * **Clean-label attack setting:** The clean label setting was configured at a low ratio because clean label attacks only poison samples within the target class. In CIFAR-10, each class comprises 10% of the overall dataset, which inherently limits the extent of poisoning achievable in this class-specific context. Therefore, a high poisoning ratio is not realistic under these constraints.
> * **Impact of low ratio in real-world scenarios:** To ensure the validity of our findings in practical applications, we have conducted additional tests under low poisoning ratios, as detailed in the Appendix Figure 9 (see Appendix C.2). These tests demonstrate that even with a low poisoning ratio, our method, AGPD, remains effective in detecting poisoned samples.
> * **Performance of gradient-based detection:** Our experiments show that AGPD outperforms feature-based methods such as AC and SCAn, particularly at low poisoning ratios. The gradient approach consistently provides better and more stable detection performance, highlighting its superiority in these scenarios.

---

> > ### Comment · Reviewer_pwS4 · 2024-11-25
> > **Response**
> >
> > I have carefully read the authors’ rebuttal and the other reviewers’ comments.  I raise the score to 6, but I still have concerns about the experiments compared to other baselines in the main paper.

---

> ### Author Response · Authors · 2024-11-25
> **Appreciation for Your Feedback**
>
> Dear Reviewer pwS4,
>
> Thanks for your positive feedback and valuable suggestions.  We will incorporate all experiments during rebuttal into the final version of the paper. Furthermore,  our code, including all the experimental details, will be released to ensure the reproducibility of all results.
>
> We sincerely appreciate any further constructive feedback you might provide to enhance our submission. Thank you once again for your thorough review and your positive evaluation.
>
> Best regards,
> Authors

---

### Official Review · Reviewer_bpNy · 2024-11-08

**Soundness:** 3
**Presentation:** 3
**Contribution:** 3
**Rating:** 6
**Confidence:** 3

**Summary:**

The authors propose to detect poisoned examples from an activation-gradient circular distribution perspective. The authors draw two novel observations by studying the GCD of poisoned/clean examples, based on which they introduce an algorithm named AGPD to do backdoor example detection.

**Strengths:**

(1) Understanding the activation gradient of poisoned examples with circular distribution is interesting. The two observations of GCD are novel.

(2) The algorithm shows strong performance within the evaluation setting of this paper.

(3) The paper is overall well written and the logic is easy to follow. The empirical experiments are mostly comprehensive and valid.

**Weaknesses:**

(1) My main concern about the proposed algorithm is its scalability. It requires training on the entire dataset to determine the detection state, which might be too expensive given web-scale data and the extreme probability of the data being poisoned. Also, it takes nearly $O(n2) $ to compute the GCD, which is infeasible for large datasets.

(2) The defense algorithm appears to be highly correlated with the specific class labels, potentially limiting its applicability. In the real world, the attackers can adopt a different set of labels than the defenders. The attackers could label the dataset more fine-grained/coarse-grained than the defender. (e.g. Imagine there is a dataset consisting of snacks and drinks and the target class of the attacker is Coca-Cola. However, the defenders just want to classify the images as soft drinks no matter whether it is Coca-Cola or Pepsi. Or the reverse situation can also happen.) Will the observations about GCD hold under such misalignment?

(3) Table 7 is not organized well, please have a double-check.

**Questions:**

Please see the weakness.

---

> ### Author Response · Authors · 2024-11-22
> **Response to Reviewer bpNy**
>
> Dear Reviewer bpNy,
>
> We sincerely appreciate the reviewer for carefully reading our paper. We are encouraged by the positive comments.
>
> **Q1. The scalability of AGPD when handling large datasets.**
>
> **R1:** Thanks for this meaningful suggestion. We would like to address this scalability concern from the following two aspects.
> * **The role of model training in poisoned sample detection (PSD):**
>     * To the best of our knowledge, **model training is widely adopted by most existing PSD methods** (including all compared methods in this work and our AGPD method), as they aim at extracting some discriminative features from the training process or the trained model to distinguish poisoned from clean samples, such as training loss (used by ABL), activation (used by AC) or activation gradient (used by AGPD). Considering that PSD serves as a data preprocessing step for the final goal of training a secure model, model training based PSD will double the overall cost. We believe that *if the defender/trainer can afford the training cost or has the sufficient training resources, the extra double cost from PSD doesn't seem unacceptable, especially considering the security benefit*.
>     * **However, we strongly agree with the reviewer that it is valuable to develop more efficient training-tree PSD methods**, to break out of the current mainstream model training based paradigm. Although it is out of the scope of this work, we think that sequential detection or utilizing some pre-trained models may be feasible solutions, and we plan to explore them in future work.
>
> * **The computational complexity of computing GCD in AGPD is linear of $N$ ($i.e.$, $O(N)$), rather than $O(N^2)$.**
>     * Firstly, the cost of Eq. (1) is $O(C^{(l)} \times H^{(l)} \times W^{(l)})$. Then, the cost of computing the angle of one sample (see Eq. (2)) is $O(2 \times C^{(l)} \times H^{(l)} \times W^{(l)})$. Thus, the cost of computing GCD (*i.e.*, computing the angles of all samples) is $O(2 \times C^{(l)} \times H^{(l)} \times W^{(l)} \times N)$. Since $N$ is often much larger than $C^{(l)}, H^{(l)}, W^{(l)}$, we can claim that **the cost of computing GCD is $O(N)$**.
>     * **The cost of AGPD (excluding the model training cost) is $O((F + B + D)N)$**, with $F, B, D$ indicating the cost of one forward pass, the cost of one backward pass along the trained model, and the feature dimension, respectively, as shown in Table 13 (see Appendix D.4). This cost is at the similar level with most compared methods. Besides, compared with model training, which cost is $O(T \times (F + B) \times N)$ with $T$ indicating the training epochs, the additional cost of AGPD is negligible.
>
> **In summary**, we believe that the cost of the proposed APGD method is acceptable, and the scalability should not be a big issue.
>
> **Q2.The defense algorithm appears to be highly correlated with the specific class labels. Will the observations about GCD hold under different set of labels?**
>
> **R2:** Thank you for your suggestion. We are currently focusing on closed-set image classification. **In backdoor attacks, the target label must be included in the model training so that samples containing the trigger are predicted as the target label.** The experimental scenario you mentioned pertains to an open-set problem, which is not included in the current backdoor attack settings. It is a challenging issue and needs future research.
>
> **Q3. Table 7 is not organized well.**
>
> **R3:** Thanks for your ward reminding. We have amended the table format in the revised version.

---

> > ### Comment · Reviewer_bpNy · 2024-11-23
> > **Thanks for the rebuttal**
> >
> > I appreciate the authors' detailed responses and all my concerns are addressed. Therefore, I have adjusted my rating.

---

> ### Author Response · Authors · 2024-11-23
> **Appreciation for Your Feedback**
>
> Dear Reviewer bpNy,
>
> Thank you for your positive feedback on our paper.
>
> The discussions we had with you have greatly improved our explanation of the concepts within the paper. The valuable insights you provided have opened up new perspectives for exploring the current issues, and we will incorporate your suggestions into our revised version.
>
> Thank you once again for taking the time to review our paper.
>
> Best regards，
>
> Authors

---

### Official Review · Reviewer_nov3 · 2024-11-11

**Soundness:** 3
**Presentation:** 3
**Contribution:** 3
**Rating:** 6
**Confidence:** 4

**Summary:**

The authors present an approach for backdoor sample detection based on the angle of the activation gradients between samples and a reference clean sample for each class.
The paper observes that, within the target class, the activation gradient angles exhibit greater dispersion compared to samples from the clean class. Furthermore, the distribution of angles of backdoor and clean samples within the target class are somewhat distinct, providing a basis for differentiating between clean and poisoned samples.

Building on this observation, the authors introduce the concept of the Gradient Circular Distribution (GCD), a distribution of angles between the activation gradient of samples and that of a reference sample. They propose two key metrics based on this distribution:
CVBT: A metric designed to measure the dispersion of the GCD.
Sample Closeness: A metric that evaluates how close a given sample is to  reference clean sample.

These metrics are then utilized to develop a filtering algorithm, which first identifies the target class and subsequently filters out the poisoned samples. The paper validates the proposed method through experiments on a range of backdoor attack scenarios.

**Strengths:**

1. The paper identifies a previously unexplored phenomenon in backdoor samples, highlighting greater dispersion in activation gradient angles within the target class compared to clean samples.
2. The authors provide extensive experiments on various backdoor attacks across two architectures (PreAct ResNet18 and VGG-18 BN),
3. Some initial analysis on adaptive attacks is presented, though further exploration could strengthen the findings.
4. The writing is clear and well-structured, making the main arguments easy to follow.

**Weaknesses:**

1. I am giving low scores to soundness due to the reported  low performance of ASSET, AC. The performance in the original paper of ASSET is close to 90% for all attacks. Additionally, 0 F1 scores for AC also does not look right. For example, in the aforementioned paper, the average TPR of AC is around 50%.

2.  Please provide sensitivity of various thresholds like $\tau_z$ , $w, \beta$ (Stage 3). What is the rationale behind the choice of $\tau_z$ ?

3.  Please comment on or provide results for these additonal adaptive attacks:
a. Can an adversary add  perturbations to the clean samples (different perturbations to different samples) such that their dispersion increases ?
b. Given a clean set with 100 images, the adversary adds the same noise to 50 images and keeps the other images the same. Will the  dispersion increases of clean sample increase ?
c. The backdoor trigger is optimized such that the dispersion of the target class decreases.

**Questions:**

1. Can you please elaborate on this : "It is inspired by phenomenon that a backdoored model tends to map both poisoned and clean samples within the target class to similar areas in its activation space" ? - I believe this is true only for self-supervised training, whereas the current paper focuses on supervised training.

2. At which h,w is the activation getting sliced for calculating the activation gradient ? Is it the same for all layers considered ? Why ?

3. Can we simply use the difference in the angles from 0 in the distribution $\rho$ instead of the sample closeness metric ? Why do we need to use it ?

---

> ### Author Response · Authors · 2024-11-22
> **Response to Reviewer nov3 (Part 1/4)**
>
> Dear Reviewer nov3,
>
> Thanks a lot for your valuable time and constructive comments.
>
> **Q1. The reasons for low performance of ASSET and AC.**
>
> **R1:** Thanks for your thoughtful review. Actually, we also noticed the unusual performance of ASSET and AC, and gave brief analysis in the original manuscript. We would like to take this chance to present more in-depth analysis and explanations, from the following aspects:
>
> * **Correctness and fairness of our implementations and evaluations:** As demonstrated in Line 290, all of our experiments setting, covering all backdoor attack and detection methods (also including ASSET, AC and our method) are conducted based on BackdoorBench [1], which is a widely recognized benchmark in backdoor learning community and has been used as the experimental platform in many recent works (*e.g.*, [2, 3, 4, 5]). It ensures the correctness and fairness of our implementations and evaluations.
>
> * **Analysis of ASSET:**
>   * **Detection mechanism of ASSET:** ASSET's detection mechanism involves dynamically learning a classifier to distinguish between the features of clean and poisoned samples.
>   * **Reason of the reported poor detection performance of ASSET:** The differing performance of ASSET in our experiments compared to the original paper can be attributed to the use of different model architectures. We employed Preact-ResNet18 in our evaluation. The distinct features of poisoned and clean samples in different architectures can significantly impact ASSET.
>   * **Additional evaluation of ASSET on other model architecture:** In **Table 1**, we provide the detection results of ASSET and AGPD using ResNet-18, which corresponds to Table 14 in Appendix. Notably, ASSET's detection performance shows improvement. However, there is still considerable variability across different attack scenarios. In contrast, AGPD demonstrates consistently better detection results than ASSET, suggesting that AGPD is a more stable detection across various models and attacks.
>
> * **Analysis of AC:**
>   * **Detection mechanism of AC:** AC's detection mechanism involves conducting cluster analysis within each class. It identifies potentially poisoned clusters based on the silhouette value and the relative size of clusters. Specifically, if a cluster's relative size is below a certain threshold and its silhouette score exceeds a silhouette threshold, the cluster is classified as poisoned (Line 272-275 in the official version [6]). This method ensures that only clusters with few members and strong internal cohesion are flagged as suspicious.
>   * **Reason of the reported poor detection performance of AC:** In our experiments, we used a poisoning rate of 10%, which led to a situation where the clean samples of the target class (fewer than 5000) are similar with poisoned samples (5000). This scenario causes AC to conclude that there are no poisoned samples within the target class because the relative sizes of clusters do not fall below the size threshold, resulting in both TPR and FPR being low.
>   * **Additional evaluation of AC in the case of low poisoning ratio:** In **Table 2**, we present the detection performance of AC with a 5% poisoning rate, corresponding to Fig. 9 in Appendix. Under these conditions, AC achieves a high TPR, demonstrating its effectiveness when the proportion of poisoned samples is more manageable.
>
> **Table 1: Performance of Asset and AGPD under poisoning ratio=10\% on CIFAR-10 and ResNet-18.**
>
> ||Asset||AGPD||
> -|-|-|-|-
> |TPR|FPR|TPR|FPR
> BadNets|90.70%|0.18%|95.94%|0.28%
> Blended|99.90%|9.43%|100.00%|3.94%
> WaNet|1.88%|1.69%|92.12%|0.36%
> Input-Aware|0.25%|0.22%|99.61%|1.46%
> TaCT|100.00%|0.00%|100.00%|3.52%
> Adap-Blend|54.58%|38.90%|88.84%|0.25%
>
> **Table 2: Performance of AC under poisoning ratio=5\%.**
>
> ||TPR|FPR|
> -|-|-
> BadNet|92.64%|0.05%
> Blended|99.64%|0.00%
> LF|94.84%|0.00%
> SSBA|98.28%|0.03%
> WaNet|92.70%|3.42%
> Input-Aware|99.36%|0.22%
> TaCT|99.96%|3.12%
> Adap-Blend|0.00%|0.00%
>
> References:
>
> [1] Wu B, Chen H, Zhang M, et al. Backdoorbench: A comprehensive benchmark of backdoor learning[J]. Advances in Neural Information Processing Systems, 2022, 35: 10546-10559.
>
> [2] Guan J, Liang J, He R. Backdoor Defense via Test-Time Detecting and Repairing[C]//Proceedings of the IEEE/CVF Conference on Computer Vision and Pattern Recognition. 2024: 24564-24573.
>
> [3] Min R, Qin Z, Shen L, et al. Towards stable backdoor purification through feature shift tuning[J]. Advances in Neural Information Processing Systems, 2024, 36.
>
> [4] Lin W, Liu L, Wei S, et al. Unveiling and Mitigating Backdoor Vulnerabilities based on Unlearning Weight Changes and Backdoor Activeness[J]. arXiv preprint arXiv:2405.20291, 2024.
>
> [5] Xian X, Wang G, Srinivasa J, et al. A unified detection framework for inference-stage backdoor defenses[J]. Advances in Neural Information Processing Systems, 2023, 36: 7867-7894.
>
> [6]https://github.com/Trusted-AI/adversarial-robustness-toolbox/blob/main/art/defences/detector/poison/clustering_analyzer.py

---

> ### Author Response · Authors · 2024-11-22
> **Response to Reviewer nov3 (Part 2/4)**
>
> **Q2. Sensitivity Analysis for thresholds in Stage 3 and the rationale for the choice of $\tau_z$**.
>
> **R2:** Thank you for this insightful suggestion. We would like to address your comments by discussing three main points:
>
> * **Mechanism of Stage 3:** Stage 3 works by first quickly detecting a large number of poisoned samples, as shown by the steep drop in the JS divergence curve in Figure 13. Over time, it starts identifying fewer, harder-to-detect poisoned samples. Eventually, it detects many clean samples, which marks the end of this stage. This process helps ensure a smooth transition from detecting poisoned to clean samples.
> * **Sensitivity of window size and $\beta$:** We tested how changes in window size and beta affect detection performance across different attacks on **Table 3** and **Table 4**. Our findings show that within a certain range, both parameters maintain strong detection performance, making our approach reliable under various conditions.
> * **Impact of window size and $\beta$ on detection:** A small window size might cause the detection process to stop too early due to fluctuations when a few poisoned samples are detected. A large window size might include clean samples, increasing the mean JS divergence and causing an early stop. For $\beta$, a small value might focus on the later stages with high variance, while a large beta might increase the false positive rate by selecting stable stages. Balancing these parameters is crucial for optimal detection.
>
> **Table 3: Performance of AGPD with different window size under poisoning ratio=10\% on CIFAR-10 and PreactResNet-18 .**
>
> ||2||4||6||8||10||12||14||16||18||20||
> -|-|-|-|-|-|-|-|-|-|-|-|-|-|-|-|-|-|-|-|-
> Attack|TPR|FPR|TPR|FPR|TPR|FPR|TPR|FPR|TPR|FPR|TPR|FPR|TPR|FPR|TPR|FPR|TPR|FPR|TPR|FPR
> BadNets|98.3|0.96|90.06|0.03|90.06|0.03|90.06|0.03|90.06|0.03|90.06|0.03|90.06|0.03|90.06|0.03|90.06|0.03|85.36|0
> Blended|93.08|2.11|93.36|0.77|94.74|1.1|91.7|0.14|94.65|0.175|90.89|0.13|93.8|0.315|93.33|0.305|91.45|0.145|90.89|0.13
> LF|99.32|0|99.28|0|99.18|0|99.04|0|99.8|0.07|99.8|0.07|99.8|0.07|99.8|0.07|98.24|0|97.86|0
> SIG|99.76|3.17|100|0.04|100|0.04|100|0.04|100|0.04|98.84|3.83|98.84|3.99|98.84|3.57|98.2|3.17|98.84|3.17
> SSBA|99.16|0.02|99.04|0.02|99.04|0.02|98.88|0.01|99.62|0.04|98.88|0.01|99.62|0.04|98.34|0.01|97.9|0.01|97.9|0.01
> TACT|99.9|0.01|99.8|0.01|99.88|0.01|99.88|0.01|100|0.07|100|0.07|100|0.07|99.18|0|99.18|0|99.18|0
> WANet|95.18|0.09|95.18|0.09|94.43|0.05|93.58|0.04|97.8|0.31|91.87|0.02|91.02|0.02|91.87|0.02|89.55|0.02|88.01|0.01
>
> **Table 4: Performance of AGPD with different $\beta$ under poisoning ratio=10\% on CIFAR-10 and PreactResNet-18 .**
>
> ||1||3||5||7||9||11||13||
> -|-|-|-|-|-|-|-|-|-|-|-|-|-|-
> Attack|TPR|FPR|TPR|FPR|TPR|FPR|TPR|FPR|TPR|FPR|TPR|FPR|TPR|FPR
> BadNets|90.06|0.03|90.06|0.03|90.06|0.03|90.06|0.03|90.06|0.03|90.06|0.03|90.06|0.03
> Blended|94.65|0.175|94.65|0.175|94.65|0.175|90.74|0.115|90.74|0.115|90.74|0.115|90.74|0.115
> LF|99.8|0.07|99.8|0.07|99.8|0.07|99.8|0.07|99.8|0.07|99.8|0.07|99.8|0.07
> SIG|100|0.04|100|0.04|100|0.04|100|0.04|99.44|3.99|99.44|3.99|99.44|3.99
> SSBA|99.62|0.04|99.62|0.04|99.62|0.04|99.62|0.04|99.62|0.04|99.62|0.04|99.62|0.04
> TACT|100|0.07|100|0.07|100|0.07|100|0.07|100|0.07|100|0.07|100|0.07
> WANet|97.8|0.31|97.8|0.31|97.8|0.31|97.8|0.31|97.8|0.31|97.8|0.31|97.8|0.31
>
> * **Guideline for setting $\tau_z$:** We recommend referring to the work [1], which provides guidelines on determining its value which is used to detect outliers from the perspective of hypothesis testing.
>   * In statistical hypothesis testing, **the significance level**, often denoted by $\alpha$, is a critical concept that represents the probability of committing a Type I error—rejecting a true null hypothesis. When using a Z-score to test hypotheses, this threshold corresponds to a specific value on the standard normal distribution.
>   * **The Z-score threshold** is determined such that the probability of observing a value as extreme, or more extreme, equal to the significance level $\alpha$.
>   * **In AGPD**, $\tau_z$ is set to $e^2$, aligning with a very stringent confidence level of approximately $\text{1e-13}$. This setup ensures the target class identified are exceptionally rare under the null hypothesis. This threshold is much stricter than typical significance levels, reducing the probability of false positives of target label to an extreme degree.
>
> References:
>
> [1] Boris Iglewicz and David C Hoaglin. Volume 16: how to detect and handle outliers. Quality Press, 1993.

---

> ### Author Response · Authors · 2024-11-22
> **Response to Reviewer nov3 (Part 3/4)**
>
> **Q3. The results and analysis for adaptive attacks.**
>
> **R3:** Thank you for posing this very insightful suggestion to enhance our approach to adaptive attacks.
>
> * **Adding noise to samples**
>   * **Experimental setup:** The poisoned dataset consists of 10% poisoned samples and 10% clean samples with added noise. In these clean samples, noise is inserted into the images of the blending ratio $0.2$ without altering the labels. There are two modes of adding noise: in the **Random mode**, all noise is generated randomly; in the **Fixed mode**, half of the noise is fixed, and the other half is generated randomly. We evaluated the effectiveness of AGPD in detecting various attacks on the CIFAR-10 dataset using the PreactResNet-18 architecture.
>   * **Samples with noise have no impact on AGPD detection.** As shown in **Table 5**, AGPD maintained high TPR and relatively low FPR across different backdoor attacks and both noise-adding modes.
>   * **Samples with noise have similar Activation Gradient with clean samples.** In the revised manuscript , we present the Activation Gradient distributions of samples with random noise, clean samples, and poisoned samples as shown in Figure 17 and Figure 18 in the revised version. We observed that in both two modes, the distribution of noise samples closely resembles that of clean samples. Despite these samples with noise, they can retain their original features, resulting in similar gradients to those of clean samples. The preservation of original features in noisy samples ensures that their gradients remain unaffected, allowing AGPD to effectively detect poisoned samples without interference from noise.
> * **Optimized trigger:** This is difficult for attackers to achieve for the following reasons:
>   * **Optimizing noise in the context of backdoor attacks involves calculating the gradient of the Activation Gradient.** AGPD method leverages the Activation Gradient to effectively differentiate between clean and poisoned samples. Optimizing trigger in the context of backdoor attacks is inherently challenging due to the complexity involved in calculating the gradient of the Activation Gradient for each layer, essentially a second-order derivative, which requires high computational resources.
>   * **The objective function is dynamic.** As outlined in our main manuscript, the detection phase employs a continually updated sample closeness metric, caused by the basis point $x_n^{*}$, during the first step of Stage 3. This continual update presents a formidable challenge in optimizing the trigger, as it requires the method to be resilient against evolving detection strategies.
>
> **Table 5: Performance of AGPD against adaptive attacks with different attacks under poisoning ratio=10\% on CIFAR-10 and ResNet-18.**
>
> |         |        | TPR    | FPR   | F1     |
> | ------- | ------ | ------ | ----- | ------ |
> | BadNets | Random | 86.68% | 0.01% | 92.83% |
> |         | Fixed  | 87.24% | 0.18% | 85.42% |
> | Blended | Random | 99.88% | 0.00% | 99.93% |
> |         | Fixed  | 99.56% | 0.00% | 99.76% |
> | LF      | Random | 99.24% | 0.01% | 99.59% |
> |         | Fixed  | 99.36% | 0.01% | 99.62% |
> | SSBA    | Random | 99.42% | 0.03% | 99.56% |
> |         | Fixed  | 99.52% | 0.03% | 99.63% |
>
> **Q4. Does the statement "a backdoored model tends to map both poisoned and clean samples within the target class to similar areas in its activation space" only hold in self-supervised training?**
>
> **R4:** Thanks for your suggestion, and it gives a chance to **clarify this misunderstanding**.
>
> * **In supervised learning**, as poisoned and clean samples within the target class share the same label, they will be guided to similar areas in its activation space through model training. This can be observed via the t-SNE visualizations of the activations in several backdoored models (see Figure 16), where poisoned samples (black points) are often more close to the target clean samples (red points) than other classes' samples (other colors).
> * **In self-supervised learning**, as the labels are not used, the samples with similar visual appearances will be close to each other in the activation space. Thus, since poisoned samples and the target clean samples are often significantly different on visual appearance, they will be mapped to separated areas. This phenomenon has been observed in DBD [1].
>
> References:
>
> [1] Kunzhe Huang, Yiming Li, Baoyuan Wu, Zhan Qin, and Kui Ren. Backdoor defense via decoupling the training process. In ICLR, 2022.

---

> ### Author Response · Authors · 2024-11-22
> **Response to Reviewer nov3 (Part 4/4)**
>
> **Q5. At which $h$,$w$ is the activation getting sliced for calculating the activation gradient? And is it the same for all layers considered?**
>
> **R5:** Thanks. **AGPD uses the channel-wise activation gradient.** The following Equation, which defines the Activation Gradient in our main manuscript, is computed by averaging over height $h$ and width $w$ to obtain the channel-wise Activation Gradient, and this is consistent across all layers. We do not need to take specific slices of $h$ or $w$. **Since poisoned samples and clean samples originate from different classes, even if their features are located in the target class space, there will be significant differences in the Activation Gradient**
> $$
> \boldsymbol{g}_{\boldsymbol{w}}^{(l)}(\boldsymbol{x}, y) = \frac{1}{H^{(l)}W^{(l)}}\sum _ {h=1}^{H^{(l)}}\sum _ {w=1}^{W^{(l)}}\frac{\partial [f _ {\boldsymbol{w}}(\boldsymbol{x})]_y}{\partial{[\boldsymbol{h}^{(l)} _ {\boldsymbol{x}}] _ {:,h,w}}} \in\mathbb{R}^{C^{(l)}},
> $$
>
> **Q6. The reason why we need sample closeness metric rather than using the angles from 0.**
>
> **R6:** Thanks for this insightful comment! Sample closeness metric incorporates two design concepts: the use of normalization and the application of cosine distance.
>
> * **Normalization is essential.** Since the gradient differences vary across different attacks, we use the sample closeness metric by normalizing the cosine distance rather than using the cosine distance directly to distinguish poisoned samples.
> * **Using $\theta$ is not superior to cosine distance.** In **Table 6**, when we replaced $1-\cos(\theta)$ in the sample closeness metric with $\theta$, there was no significant change in the TPR across different attacks. However, the FPR significantly increased under certain attacks. Cosine distance is an intuitive criterion for measuring angular differences.
>
> **Table 6: Performance (%) of AGPD-Theta and AGPD with different attacks under poisoning ratio=10\% on CIFAR-10 and PreactResNet-18.**
>
> |             | AGPD-Theta |        |        | AGPD    |       |        |
> | ----------- | ---------- | ------ | ------ | ------- | ----- | ------ |
> |             | TPR        | FPR    | F1     | TPR     | FPR   | F1     |
> | BadNet      | 87.14%     | 0.16%  | 92.40% | 90.06%  | 0.03% | 94.65% |
> | Blended     | 99.84%     | 0.00%  | 99.91% | 99.98%  | 0.02% | 99.88% |
> | LF          | 99.50%     | 0.01%  | 99.69% | 99.80%  | 0.07% | 99.60% |
> | SSBA        | 99.44%     | 0.02%  | 99.64% | 99.62%  | 0.04% | 99.63% |
> | WaNet       | 94.77%     | 26.88% | 41.69% | 97.80%  | 0.31% | 97.40% |
> | Input-Aware | 97.59%     | 18.08% | 52.42% | 88.25%  | 1.13% | 88.61% |
> | TaCT        | 99.68%     | 0.00%  | 99.82% | 100.00% | 0.07% | 99.68% |
> | Adap-Blend  | 90.00%     | 1.07%  | 90.15% | 89.32%  | 0.33% | 92.89% |

---

> ### Author Response · Authors · 2024-11-26
> **Official Comment by Authors**
>
> Dear Reviewer nov3,
>
> We would like to express our sincere gratitude for your valuable insights and suggestions on our work.
>
> We have tried our best to address the concerns and queries you raised during the rebuttal process. However, we would greatly appreciate knowing whether our response has effectively resolved your doubts. Your feedback will be instrumental in improving the quality of our work. As the end of the discussion period is approaching, we eagerly await your reply before the end.
>
> Best regards,
>
> Authors

---

> > ### Comment · Reviewer_nov3 · 2024-11-26
> >
> > I thank the authors for their effort to address my concerns and comments.
> > I am convinced about the effectiveness of their approach due to the additional experiments, i.e. robustness to hyperparameters and adaptive attacks.
> > I have raised my score accordingly.
> >
> > If the authors are confident about the performance of ASSET deteriorating so much due to the presence of a different network, I suggest the authors to kindly mention it in their manuscript for future works.

---

> > > ### Author Response · Authors · 2024-11-27
> > > **Appreciation for Your Feedback**
> > >
> > > Dear Reviewer nov3,
> > >
> > > Thank you for your positive feedback and valuable suggestions. We greatly appreciate your feedback, which allows us to explain the robustness of our method further. We will incorporate the current experimental analysis into the final version of the manuscript.
> > >
> > > Once again, thank you for your thorough review and positive evaluation.
> > >
> > > Best regards,
> > >
> > > Authors

---

### Comment · Area_Chair_nZAG · 2024-11-24

Dear reviewers,

Thanks for serving as a reviewer. As the discussion period comes to a close and the authors have submitted their rebuttals, I kindly ask you to take a moment to review them and provide any final comments.

If you have already updated your comments, please disregard this message.

Thank you once again for your dedication to the OpenReview process.

Best,

Area Chair

---

### Meta-Review · Area_Chair_nZAG · 2024-12-21

**Metareview:**

The paper proposed a new apporach for detecting poisoned samples called Activation Gradient-Based Poisoned Sample Detection (AGPD). It leveerages a novel metric called Gradient Circular Distribution to distinguish clean and poisoned samples. The empirical results also demonstrate its effectiveness.

Strength:

1. Insightful view on understanding the activated gradient of poisoned examples with distribution.

2. The algorithm shows strong performance within the evaluation setting of this paper.

3. Well written.

Weakness:

1. Only explores the backdoor attacks on Pre-Act ResNet 18 and VGG, whether the proposed feature still exists in Transformers or other networks are unexplored.

All the reviewers thoughts positive towards this paper, therefore I recommend to accept it as a poster.

**Additional Comments On Reviewer Discussion:**

The reviewers and authors discussed about the baseline attacks and detection methods, etc. The discussions further improve the paper.

---

### Decision · Program_Chairs · 2025-01-22

Accept (Poster)